# DISTRIBUTION-CALIBRATED INFERENCE TIME COMPUTE FOR THINKING LLM-AS-A-JUDGE

## ABSTRACT

Thinking Large Language Models (LLMs) used as judges for pairwise preferences remain noisy at the single-sample level, and common aggregation rules (majority vote, soft self-consistency, or instruction-based self-aggregation) are inconsistent when ties are allowed. We study *inference-time compute* (ITC) for evaluators that generate $n$ independent thinking–rating samples per item, and propose a principled, distribution-calibrated aggregation scheme. Our method models three-way preferences with a Bradley–Terry-Davidson formulation on rating counts, leveraging both *polarity* (margin among non-ties) and *decisiveness* (non-tie rate) to distinguish narrow margins from strong consensus. Across various evaluation benchmarks, our approach consistently reduces MAE and increases pairwise accuracy versus standard baselines, and when evaluated against human-consensus meta-labels, matches or exceeds individual human raters. These results show that carefully allocating ITC and aggregating with distribution-aware methods turns noisy individual model judgments into reliable ratings for evaluation.

## 1 INTRODUCTION

Thinking large language models (LLMs) are increasingly being employed as automated judges for evaluating the output of other generative systems, a paradigm known as "Thinking-LLM-as-a-Judge" (Saha et al., 2025). This approach offers a scalable and cost-effective alternative to human evaluation, which is often slow and expensive. To mitigate the inherent stochasticity and noise of single-pass judgments, a common strategy is to leverage inference-time compute (ITC) Snell et al. (2024) by generating multiple independent reasoning and rating samples for each item being evaluated. However, the reliability of the final judgment hinges critically on how these multiple outputs are aggregated.

Current aggregation methods, such as majority voting (Self-Consistency (Wang et al., 2023b)) or heuristics based on model confidence scores or LLM generated aggregators, are often brittle and statistically suboptimal. These approaches are particularly fragile in the presence of ties. For instance, a simple majority vote cannot distinguish between a narrow 5-to-4 decision and a decisive 9-to-0 consensus, discarding valuable information about the strength of evidence contained within the full distribution of votes. This insensitivity to evidential strength leads to less reliable and robust evaluations.

In this work, we argue that the aggregation step is not an afterthought but a critical component for effectively utilizing ITC. We propose a principled, Distribution-Calibrated Aggregation scheme that moves beyond simple vote-counting. Our method operates directly on the full counts of positive, negative, and tie votes, preserving the full signal in the sample distribution. Specifically, we model the three-way preference outcomes using a Bradley-Terry-Davidson (Davidson, 1970) formulation, which explicitly parametrizes both the preference margin and the global propensity for ties. By estimating parameters via maximum likelihood on a small calibration set and then using the Mean Absolute Error (MAE) Bayes action at inference, our approach stays aligned with the evaluation metric while leveraging a well-behaved probabilistic fit, avoiding loss–metric mismatch and yielding more accurate judgments. Conceptually, this calibration step modifies the decision boundary compared to a simple majority voting as demonstrated in Figure 1.

We conduct extensive experiments on a diverse set of benchmarks, including machine translation evaluation (WMT23) (Song et al., 2025) and reward model assessment (Reward Bench 2) (Malik

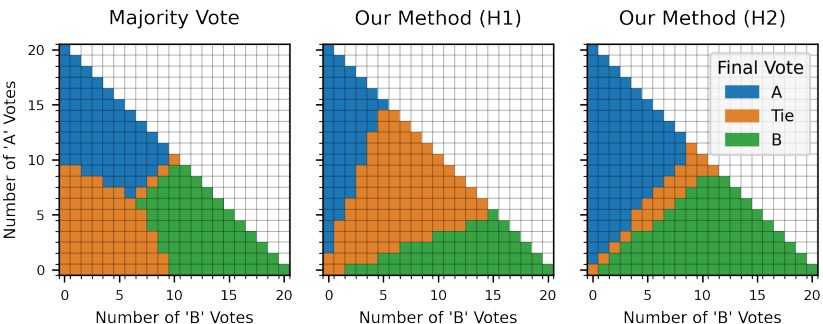

Figure 1: Behavior of Different Aggregation Methods with 20 Votes. Our proposed method's behavior is shown using two different hyperparameters. The number of 'Tie' votes is computed as 20 - (# of A votes) - (# of B votes)

et al., 2025). Our results show that our distribution-calibrated approach considerably outperforms a suite of strong self-consistency baselines. By carefully modeling the entire vote distribution, our method turns noisy individual model judgments into more reliable ratings, matching or exceeding the performance of individual human raters when evaluated against a human-consensus gold standard.

**Contributions**: Our main contributions are threefold: (1) We show that the existing aggregation methods for inference time compute for LLM judges are suboptimal and that a carefully designed aggregation approach is critical. (2) We propose an Expected Risk Minimization (ERM)-based Bradley–Terry–Davidson aggregation fit on a small calibration set, and show that it consistently outperforms existing aggregation methods across different tasks in both reward benchmarks and MT. (3) For MT in particular, we adopt a consensus-based meta-evaluation to form higher-fidelity ground truths where labels are noisy, enabling fair comparison to human raters and revealing regimes where LLM judges approach "super-human" evaluation quality.

## 2 RELATED WORK

**LLM-as-a-Judge**: Recently, Large Language Models (LLMs) have achieved remarkable success when deployed as "judges" (Zheng et al., 2023b) to evaluate generated text, offering a scalable alternative to traditional metrics (Gu et al., 2025). This paradigm has demonstrated high correlation with human judgments across diverse domains. Approaches vary: some prompt general-purpose LLMs directly (e.g., G-Eval (Liu et al., 2023); JudgeLM (Zhu et al., 2025)), while others fine-tune specialized models optimized for evaluation tasks (e.g., Prometheus (Kim et al., 2023); Auto-J (Li et al., 2023)). While powerful, these LLM-based approaches face significant challenges, including sensitivity to prompt design (Gu et al., 2025) and inherent biases, such as positional bias (favoring a specific candidate order) or verbosity bias (preferring longer outputs) (Wang et al., 2023a). Moreover, LLM judges exhibit variability in their decision-making, with some models being more aggressive than others in breaking subtle distinctions or ties (Zheng et al., 2023b). Our work focuses on mitigating this noise and improving the reliability of judgments through a principled aggregation.

**Thinking in Language Models for Evaluation.** The reliability of LLM judgments is often enhanced when the model is prompted to generate intermediate reasoning steps before emitting a final verdict, a technique popularized by Chain-of-Thought (CoT) prompting (Wei et al., 2022). In the context of evaluation, this "thinking" process allows the model to articulate the criteria for judgment and justify its decision, leading to the "Thinking-LLM-as-a-Judge" paradigm (Saha et al., 2025). This explicit reasoning not only improves the accuracy of the judgments (Zhang et al., 2025) but also increases their interpretability. Our work leverages the generation of these independent thinking traces and investigates how to best aggregate the resulting rating samples.

**Inference Time Compute and Sample Aggregation**: Multiple strategies have been proposed that leverage inference time compute (Liu et al., 2025) When multiple samples are generated using ITC, an aggregation strategy is required. Self-Consistency (SC) (Wang et al., 2023b) aggregates multiple outputs using majority voting. Several variants incorporate confidence signals. Soft Self-consistency

(Soft-SC) (Wang et al., 2024) picks the minimum, mean, or product of confidence scores of items in each category. Confidence-Informed Self-Consistency (CI-SC) (Taubenfeld et al., 2025) computes a weighted majority vote based on confidence scores, which are computed as either the length-normalized probability of the sequence or via prompting an LLM. Alternatively, some methods leverage the LLM itself for aggregation. Generative Self-Aggregation (GSA) (Li et al., 2025) asks the LLM to synthesize a new response based on the context of multiple samples. Universal Self-Consistency (USC) (Chen et al., 2023) leverages the LLM to select the most consistent answer among multiple candidates. Furthermore, Singhi et al. (2025) and Zhang et al. (2025) showed that one can improve the performance of reasoning-based generative verifiers via test-time compute, particularly via majority voting.

**Generator Refinement and Verification**: A different line of work refines the generation process itself. Methods like Mirror-Consistency (Li et al., 2024), Self-Contrast (Zhang et al., 2024), and Step-Back Prompting (Zheng et al., 2023a) utilize iterative reflection or diverse perspectives to produce higher-quality samples, while Self-Check (Miao et al., 2023) employs step-wise verification to filter errors. Unlike these approaches, which focus on enhancing the generator (often incurring sequential computational costs), our work focuses on the aggregator: we accept the noisy distribution of parallel samples and apply a distribution-calibrated layer to robustly estimate the ground truth.

## 3 MOTIVATION: THE TIE DILEMMA

A critical choice when designing an LLM-as-a-Judge for pairwise comparisons (Zheng et al., 2023b) is whether to allow the judge to declare a tie or to force it to pick a preference. In this section, we first show that forcing the model to break ties might induce LLM biases. We then show that the tie decisions are highly sensitive to the judge parameters which requires a more robust aggregation method to mitigate.

**Ties are important to reduce LLM biases**: LLM-as-a-Judge exhibit multiple types of systematic biases (Ye et al., 2024). A well-known issue is positional bias (Shi et al., 2025), where the model's preference can be affected by the order in which responses are presented.

To quantify this, we evaluated several LLMs (qwen3-next-80b (Qwen Team, 2025), gpt-oss-120b (OpenAI, 2025), deepseek-v3.1(DeepSeek-AI, 2024) and gemini-2.5-flash (Comanici et al., 2025)) on a subset of 336 pairs of responses from the WMT23 ZH → EN dataset which we discuss in details in Section 5. The subset was limited to pairs rated as ties by humans since we are interested in studying the behavior of the LLMs around the ties boundary. We rate each pair twice by swapping the positions, thus an unbiased LLM should prefer the first and second responses on average equally. We present results in Table 1 for the two models (qwen3-next-80b and gemini-2.5-flash) that exhibited notable bias, in the forced-choice setting where a "tie" was not an option. For example, gemini-2.5-flash shows a strong 14.58% bias toward the first answer, while qwen3-next-80b exhibits an 8.04% bias toward the second. Both other models, gpt-oss-120b and deepseek-v3.1, had less than 1% positional bias in this setup and thus were not included in the table.

The right side of Table 1 shows the results from the same experiment but with the prompt updated to allow ties. The introduction of this third choice dramatically reduces positional bias for both models. This demonstrates that including a tie option is not just a feature for capturing equivalence, but might be a critical mechanism for debiasing the evaluation process itself.

Table 1: Allowing a 'Tie' Option Reduces Positional Bias. The table compares preferences in a forced-choice setting against one where a 'tie' is allowed. The bias is computed as (#First - #Second) / (#First + #Second)

| Model | Forced-Choice (No Tie) | | | Tie Allowed | | | |
|---|---|---|---|---|---|---|---|
| | **First** | **Second** | **Bias** | **First** | **Second** | **Tie** | **Bias** |
| gemini-2.5-flash | 385 | 287 | 14.5% | 220 | 199 | 253 | 3.1% |
| qwen3-next-80b | 309 | 363 | -8.0% | 322 | 318 | 32 | 0.6% |

**Tie decisions are not stable**: Our work is motivated by the fact that in three-way preference tasks, the vote distribution of an LLM-as-a-judge is highly sensitive to variations in the evaluation setup.

In this section, we demonstrate empirically two major sources of variability in ratings - (1) the LLM queried and (2) the prompt template used to get the ratings. We conduct an experiment where we generated three slight variations of an evaluation prompt as shown in Appendix A. We then use each of these prompts to judge the same dataset from the previous section. As shown in Table 2, the results reveal a significant variance in the rate of ties across prompts and LLMs. For instance, using the gemini-2.5-flash model, the percentage of "Ties" votes fluctuates dramatically, ranging from a high of 37.6% with `prompt_3` to a low of 12.4% with `prompt_1`. We also observe that deepseek-v3.1 produces an average tie rate of 30.4% across all prompts, which is significantly higher than gpt-oss-120b's average of 21.8%.

Table 2: Ties Rates for Different Models and Prompts (in %)

| Model | prompt_1 | prompt_2 | prompt_3 | Model Avg |
|---|---|---|---|---|
| gpt-oss-120b | 19.3% | 24.4% | 21.6% | 21.8% |
| gemini-2.5-flash | 12.4% | 21.3% | 37.6% | 23.8% |
| deepseek-v3.1 | 28.9% | 29.6% | 32.6% | 30.4% |
| **Prompt Avg** | 20.2% | 25.1% | 30.6% | 25.3% |

This instability is a critical flaw for methods that do not calibrate for such variations since a simple change in prompt wording can fundamentally alter the tie likelihood. This underscores the need for a robust distribution-calibrated aggregation method, which can explicitly model and adapt to these shifts. Other works have investigated calibration via finetuning the model Park et al. (2024) Ye et al. (2025); in this work we focus at mitigation strategies at inference time.

## 4 DISTRIBUTION-CALIBRATED INFERENCE-TIME SAMPLE AGGREGATION

**Setting.** Given a prompt $x$ and a pair of responses $(t_1, t_2)$, our autorater queries a Thinking LLM $n$ times to obtain independent reasoning–rating tuples $\{(z_j, r_j)\}_{j=1}^n$, where $z_j$ is a thinking trace and $r_j \in \{-1, 0, +1\}$ is a discrete vote (+1: $t_1 \succ t_2$, -1: $t_2 \succ t_1$, 0: tie). Empirically, once a thinking trace $z_j$ is produced, the conditional distribution $p(r_j \mid z_j, \cdot)$ is sharply peaked (Wang et al., 2025). In addition, we do not see a high variation in the normalized probability of the thinking traces. We therefore find that log-likelihood reweighting adds little signal in practice. Instead, we operate directly on the vote counts, which preserve the strength of evidence in the sample distribution. Let

$$c^+ = \big|\{j : r_j = +1\}\big|, \quad c^- = \big|\{j : r_j = -1\}\big|, \quad c^0 = \big|\{j : r_j = 0\}\big|, \quad n = c^+ + c^- + c^0,$$

and equivalently $\mathbf{n} = (c^+, c^0, c^-)$. While majority vote (the mode of $\mathbf{n}$) is common, it is statistically suboptimal: it is highly sensitive to sampling noise and ignores evidential strength (e.g., it cannot distinguish 5–to–4 from 9–to–0). We instead aggregate via a parametric model that consumes the full count distribution and is aligned to our evaluation metric.

**Evaluation Metric.** Let $y^\star \in \{-1, 0, +1\}$ denote the ground truth and $\hat{y}$ the aggregator's decision. We evaluate with mean absolute error (MAE):

$$\text{MAE} = \frac{1}{n} \sum_{i=1}^n \ell(\hat{y}_i, y_i^\star), \qquad \ell(a, b) = |a - b|. \tag{1}$$

This ordinally-aware metric is well-suited for the ordered label set $\{-1, 0, +1\}$. Unlike standard accuracy, which treats all misclassifications uniformly, MAE scales penalties by severity: it penalizes complete preference reversals (error of magnitude 2) more heavily than tie-related disagreements (error of magnitude 1), thereby preserving the semantic hierarchy of the preference scale.

**Count-derived features from votes.** We extract two smoothed features from $\mathbf{n}$:

$$s = \tfrac{1}{2} \log \frac{c^+ + \alpha}{c^- + \alpha}, \tag{2}$$

with small $\alpha > 0$ (we use $\alpha=1$), capturing the decisive margin; and a tie-evidence feature

$$t = \log \frac{c^0 + \kappa}{n + \kappa} \leq 0, \tag{3}$$

with $\kappa > 0$ (we use $\kappa{=}1$), which increases (toward 0) as ties appear more frequently.

**A Davidson-style model with ties.** We adopt a multinomial logit model inspired by the Bradley–Terry–Davidson framework for ternary outcomes. For an item with a latent margin $u \in \mathbb{R}$ and a tie logit $\eta \in \mathbb{R}$,

$$p(+1) = \frac{e^u}{Z}, \quad p(-1) = \frac{e^{-u}}{Z}, \quad p(0) = \frac{e^\eta}{Z}, \quad Z = e^u + e^{-u} + e^\eta. \tag{4}$$

We link features to scores linearly and *jointly* model both decisive margin and tie propensity:

$$u = \beta\, s, \qquad \eta = \eta_0 + \gamma\, t, \tag{5}$$

with parameters $\theta = (\beta, \eta_0, \gamma)$. This single specification allows a global tie baseline via $\eta_0$ and item-specific modulation via $t$ with slope $\gamma$.

**MAE-aligned decision rule.** Given $\theta$ and an input $(s, t)$, we compute probabilities via Equation 4–equation 5. The Bayes-optimal action under MAE is the label $y \in \{-1, 0, +1\}$ that minimizes the expected risk:

$$\begin{aligned}
\mathcal{R}(-1) &= p(0) + 2\, p(+1), \\
\mathcal{R}(0) &= p(+1) + p(-1), \\
\mathcal{R}(+1) &= 2\, p(-1) + p(0).
\end{aligned} \tag{6}$$

The optimal decision is therefore given by:

$$\hat{y} = \arg \min_{y \in \{-1, 0, +1\}} \mathcal{R}(y). \tag{7}$$

**Parameter fitting via The Discrete Ranked Probability Score.** A direct approach is to minimize empirical MAE on a held-out calibration set $\mathcal{C}$:

$$\hat{\theta} \in \arg \min_\theta \; \frac{1}{|\mathcal{C}|} \sum_{i \in \mathcal{C}} \ell\big(\hat{y}_i(\theta),\, y_i^\star\big), \tag{8}$$

where $\hat{y}_i$ is obtained by computing $(u_i, \eta_i)$ from $(s_i, t_i)$, the Davidson probabilities (Equation 4), and then the MAE Bayes action (Equation 7). However, Equation 8 is ill-suited for standard gradient-based methods as predictions change only when a decision boundary is crossed.

To address this problem, we decouple the model fitting from the decision rule. We fit the probabilistic model by minimizing the *Discrete Ranked Probability Score* (DRPS), a strictly proper scoring rule designed for ordinal outcomes (Gneiting & Raftery, 2007). Let the ordered label set be $\{-1, 0, +1\}$ and define the cumulative probabilities:

$$F_{-1} \equiv \Pr(Y \le -1) = p(-1), \qquad F_0 \equiv \Pr(Y \le 0) = p(-1) + p(0). \tag{9}$$

For an observation $y^\star$, define the corresponding cumulative indicators:

$$H_{-1}(y^\star) = \mathbb{1}\{y^\star \le -1\}, \qquad H_0(y^\star) = \mathbb{1}\{y^\star \le 0\}. \tag{10}$$

where $\mathbb{1}$ denotes the indicator function. The per-item DRPS is the squared CDF discrepancy:

$$\mathrm{DRPS}\big(p(\cdot \mid s, t), y^\star\big) = \big(F_{-1} - H_{-1}(y^\star)\big)^2 + \big(F_0 - H_0(y^\star)\big)^2. \tag{11}$$

We then estimate the parameters $\theta$ via empirical risk minimization on the calibration set $\mathcal{C}$:

$$\hat{\theta} \in \arg \min_\theta \; \frac{1}{|\mathcal{C}|} \sum_{i \in \mathcal{C}} \mathrm{DRPS}(p_\theta(\cdot \mid s_i, t_i),\, y_i^\star). \tag{12}$$

This approach is preferable to direct MAE minimization for three reasons: (i) **Fisher Consistency.** As a *strictly proper scoring rule* for ordinal outcomes, DRPS is uniquely minimized by the true data-generating distribution (Gneiting & Raftery, 2007). This guarantees Fisher consistency—recovery of the true parameters $\theta$ in the population limit. (ii) **Alignment with MAE Decision Rule.** Our final decision action is the MAE Bayes rule in Equation 7, which depends on well-calibrated class

---

**Algorithm 1** Inference-time aggregation with a calibrated Davidson model

---

**Require:** Calibration set $\mathcal{C}$, source query $x$, response pair $(t_1, t_2)$, sampling budget $n$, smoothing factors $(\alpha, \kappa)$.

1: **Calibrate parameters (offline, once).** For each $i \in \mathcal{C}$, tally $(c_i^+, c_i^-, c_i^0)$ and compute $s_i = \frac{1}{2} \log \frac{c_i^+ + \alpha}{c_i^- + \alpha}$ and $t_i = \log \frac{c_i^0 + \kappa}{n_i + \kappa}$ (Equation 2–equation 3). Fit $\hat{\theta} = (\hat{\beta}, \hat{\eta}_0, \hat{\gamma})$ by minimizing the empirical DRPS equation 12 with L-BFGS-B (few random restarts).

2: **Aggregate a new pair.** Query the LLM $n$ times to obtain votes $\{r_j\}_{j=1}^n \subset \{-1, 0, +1\}$; tally $(c^+, c^-, c^0)$; compute $s = \frac{1}{2} \log \frac{c^+ + \alpha}{c^- + \alpha}$ and $t = \log \frac{c^0 + \kappa}{n + \kappa}$.

3: Form $(u, \eta) = (\hat{\beta} s, \hat{\eta}_0 + \hat{\gamma} t)$ and compute $p(-1), p(0), p(+1)$ via Equation 4.

4: Compute risks $\mathcal{R}(-1), \mathcal{R}(0), \mathcal{R}(+1)$ via Equation 6.

5: **Output** $\hat{y}$ via the Bayes action Equation 7.

---

probabilities. While MAE is an ordinally-aware metric for point estimates, the DRPS is its natural generalization to probabilistic forecasts. Minimizing DRPS produces calibrated, ordinally-aware probabilities, ensuring that the downstream Bayes action equation 7 is asymptotically risk-optimal for the MAE metric. (iii) **Superior Optimization Landscape.** Unlike the non-smooth ERM–MAE objective equation 8, the DRPS objective in equation 12 is differentiable with respect to $\theta$. This enables efficient estimation using quasi-Newton methods (e.g., L-BFGS-B) under simple box constraints (Nocedal & Wright, 2006).

Hence, we fit the model by minimizing the empirical DRPS on a calibration set and apply the MAE Bayes decision rule at inference time. This two-stage procedure is summarized in Algorithm 1.

## 5 EXPERIMENTS

**Baselines**: In our experiments, we consider the following baselines:

1. Greedy decoding (GD): draws $n = 2$ samples with reversed order and a temperature of zero.
2. Few Shot (FS): draws $n = 2$ samples with reversed order with the labeled calibration set provided in the prompt as in-context examples. We use a temperature of zero.
3. Self-Consistency (SC) (Wang et al., 2023b): aggregates multiple outputs using majority voting.
4. Soft Self-Consistency (Soft-SC) (Wang et al., 2024): picks the minimum, mean, or product of confidence scores within each category.
5. Confidence-Informed Self-Consistency (CI-SC) (Taubenfeld et al., 2025): computes a weighted majority vote based on confidence scores; here we use the length-normalized probability of the sequence ($\in [0, 1]$). Alternatively, one could prompt an LLM for the confidence score (Kadavath et al., 2022), but in our experiments the LLM was almost always highly confident.
6. Generative Self-Aggregation (GSA) (Li et al., 2025): asks the LLM to synthesize a new response based on the context of multiple samples.
7. Universal Self-Consistency (USC) (Chen et al., 2023): leverages the LLM to select the most consistent answer among multiple candidates.

In both GD and FS, we aggregate the two responses using a rounded median, where a pair of $(0, 1)$ is mapped to 1. Empirically, this choice leads to better results in both cases. In other baselines, to overcome the positional bias, we draw $\frac{n}{2}$ samples in an A-then-B response order and the remaining $\frac{n}{2}$ samples via a B-then-A order. We then aggregate the entire $n$ samples. In our experiments (except for the GD and FS baselines), we use temperature sampling with a $T = 0.5$ to generate the candidates (Appendix G). For LLM aggregation methods, we use greedy decoding in the aggregation stage. All the LLM calls in this paper are done through Thinking LLMs with thinking enabled.

**Thinking Models** We consider the following Thinking LLMs: gemini-2.5-flash (Comanici & et al., 2025), qwen3-next-80b (Qwen Team, 2025), gpt-oss-120b (OpenAI, 2025).

**Benchmarks** We consider two machine translation tasks (Song et al., 2025) and six tasks from the Reward Bench 2 benchmark (Malik et al., 2025). See Appendix A for the prompts.

We use the WMT23 (Song et al., 2025) dataset and focus on two tasks for two different language pairs EN → DE and ZH → EN. For each source sentence and its two possible translations, the dataset

Table 3: MAE (lower score is better) over different tasks with different methods for $n \in \{4, 12\}$ via gemini-2.5-flash.

| Dataset | Ours | | SC | | Soft-SC | | CI-SC | | USC | | GSC | |
|---|---|---|---|---|---|---|---|---|---|---|---|---|
| | 4 | 12 | 4 | 12 | 4 | 12 | 4 | 12 | 4 | 12 | 4 | 12 |
| WMT EN → DE | **0.591** | **0.588** | 0.671 | 0.648 | 0.664 | 0.673 | 0.667 | 0.652 | 0.728 | 0.765 | 0.723 | 0.755 |
| WMT ZH → EN | **0.506** | **0.497** | 0.549 | 0.527 | 0.557 | 0.560 | 0.544 | 0.524 | 0.527 | 0.505 | 0.581 | 0.546 |
| RB2-Factuality | **0.487** | **0.451** | 0.615 | 0.647 | 0.681 | 0.711 | 0.675 | 0.670 | 0.575 | 0.573 | 0.599 | 0.591 |
| RB2-Focus | **0.332** | **0.287** | 0.394 | 0.403 | 0.397 | 0.370 | 0.415 | 0.415 | 0.424 | 0.423 | 0.439 | 0.441 |
| RB2-Math | **0.306** | **0.285** | 0.360 | 0.384 | 0.400 | 0.372 | 0.391 | 0.385 | 0.410 | 0.415 | 0.427 | 0.450 |
| RB2-Precise IF | **0.451** | **0.414** | 0.498 | 0.552 | 0.581 | 0.603 | 0.551 | 0.570 | 0.574 | 0.530 | 0.597 | 0.524 |
| RB2-Safety | **0.319** | **0.285** | 0.373 | 0.402 | 0.406 | 0.409 | 0.412 | 0.405 | 0.407 | 0.405 | 0.406 | 0.414 |
| RB2-Ties | **0.094** | **0.081** | 0.155 | 0.158 | 0.177 | 0.177 | 0.178 | 0.165 | 0.226 | 0.221 | 0.208 | 0.197 |

contains 6 multiple ratings. Three ratings were collected using a simplified side-by-side task in which raters compare two translations and assign labels $\{-1, 0, +1\}$. The other three other ratings were collected using direct assessment with MQM (Lommel et al., 2013) which we converted to a $\{-1, 0, +1\}$ by looking at the difference in absolute score. The WMT EN → DE set comprises $\sim 500$ document-level segments rated by 10 human raters, whereas the WMT ZH → EN set comprises $\sim 1,800$ sentence-level segments rated by 8 humans. We aggregate the six ratings by majority vote to obtain a consensus label, which serves as the gold standard. We selected this benchmark because it provides multiple independent human ratings per segment which allows us to benchmark our approach against individual human raters by performing leave-one-out comparisons.

The Reward Bench 2 benchmark (Malik et al., 2025) is designed for evaluating reward models across six distinct domains: Factuality, Precise Instruction Following (IF), Math, Safety, Focus, and Ties. For our evaluation, we constructed preference pairs by generating all possible pairs from each task's source dataset, which contains both accepted and rejected responses. These pairs are categorized into 'non-tie' pairs (pairing one accepted and one rejected response) and 'tie' pairs (pairing two accepted or two rejected responses). From this comprehensive set, we then sample 1000 examples for each of the six tasks to form the final benchmark. We provide a detailed breakdown of the ground truth vote distributions for each task in Appendix B.

**Meta Evaluation Metrics**: We report mean absolute error (MAE) on ordinal labels $y_i \in \{-1, 0, +1\}$ using Equation 1. We use MAE for model selection and ablations. We also report pairwise accuracy, $\text{PA} = \frac{1}{n}\sum_{i=1}^{n} \mathbf{1}[\hat{y}_i = y_i]$.

**Experimental Setup**: We randomly sample $\alpha|\mathcal{D}|$ test samples as the calibration set (for our method, and also for the FS baseline) and use the rest of the samples for evaluation (for all the methods including ours), and report the average results over 100 random calibration-evaluation splits. We use $\alpha = 5\%$ as the ratio of test samples for calibration for all the tasks. Increasing the size of the calibration set seems to slightly improve the results in some tasks, but typically this small calibration set size is sufficient for our calibration method.

**Results**: Tables 3 and 4 report MAE and pairwise accuracy for all aggregation methods using gemini-2.5-flash at $n \in \{4, 12\}$ across tasks. After scoring on 100 calibration-evaluation splits, we identify the *top cluster* using the procedure of Freitag et al. (2023): sort aggregation methods by average score and assign rank 1 to consecutive methods until we encounter the first that is significantly different from any already included method; all rank 1 methods are bolded in the tables. Significance is determined via a paired permutation test: for each pair of aggregation methods, we compare per-item outcomes on each evaluation set and obtain a $p$-value using random resampling (100 resamples per split), with $\tau = 0.05$.

Our method attains the best scores on all the datasets and sample counts. Across RB2 tasks, increasing $n$ from 4 to 12 consistently improves our method, whereas SC tends to degrade or remain flat. Other aggregation baselines vary non-monotonically with $n$ in a task-dependent manner. In the majority of tasks, we find that the evaluation performance plateaus at around $n = 12$ samples with RB2-Ties, RB2-Focus, and RB2-Precise IF showing marginal gains at $n = 20$ compared to $n = 12$.

Table 4: Pairwise accuracy (higher score is better) over different tasks with different methods for $n \in \{4, 12\}$ via gemini-2.5-flash

| Dataset | Ours | | SC | | Soft-SC | | CI-SC | | USC | | GSC | |
|---|---|---|---|---|---|---|---|---|---|---|---|---|
| | 4 | 12 | 4 | 12 | 4 | 12 | 4 | 12 | 4 | 12 | 4 | 12 |
| WMT EN → DE | **0.510** | **0.516** | 0.442 | 0.467 | 0.496 | 0.477 | 0.473 | 0.465 | 0.436 | 0.447 | 0.452 | 0.463 |
| WMT ZH → EN | **0.583** | **0.607** | 0.515 | 0.539 | 0.528 | 0.529 | 0.530 | 0.545 | 0.561 | 0.590 | 0.512 | 0.550 |
| RB2-Factuality | **0.536** | **0.566** | 0.450 | 0.424 | 0.410 | 0.399 | 0.409 | 0.411 | 0.472 | 0.475 | 0.445 | 0.461 |
| RB2-Focus | **0.685** | **0.725** | 0.629 | 0.626 | 0.636 | 0.663 | 0.616 | 0.616 | 0.604 | 0.612 | 0.601 | 0.602 |
| RB2-Math | **0.709** | **0.723** | 0.658 | 0.635 | 0.626 | 0.654 | 0.632 | 0.634 | 0.616 | 0.619 | 0.609 | 0.605 |
| RB2-Precise IF | **0.572** | **0.605** | 0.556 | 0.530 | 0.507 | 0.490 | 0.528 | 0.515 | 0.495 | 0.522 | 0.474 | 0.527 |
| RB2-Safety | **0.691** | **0.723** | 0.650 | 0.630 | 0.635 | 0.633 | 0.626 | 0.629 | 0.619 | 0.623 | 0.625 | 0.618 |
| RB2-Ties | **0.905** | **0.918** | 0.844 | 0.842 | 0.823 | 0.822 | 0.822 | 0.834 | 0.773 | 0.779 | 0.792 | 0.804 |

We compare the behavior of different aggregation methods versus $n$ over the RB2-Precises IF task in Figure 2. In this Figure, Error bars show $95\%$ confidence intervals of the mean over the 100 random calibration–evaluation splits, computed as $\bar{x} \pm 1.96\,\mathrm{SE}$ for each $n$ and method. Note that Ours is the only method that fits parameters on the calibration set every time, which injects an extra source of variability to its curve. For FS, due to its high cost (since we need to regenerate the samples for every calibration-evaluation split), we averaged the results over 10 random splits. Our method outperforms all the baselines by a large margin.

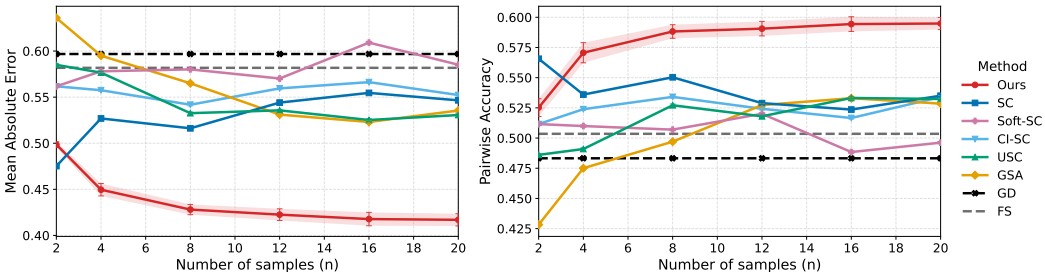

Figure 2: MAE and Pairwise Accuracy versus $n$ on RB2-Precise IF task for different methods.

For WMT ZH → EN, we conduct an additional meta evaluation comparing the ITC LLM judge to individual human raters via a leave-one-out (LOO) protocol. Given ratings from $k$ raters $R_1, \ldots, R_k$, we iteratively drop $R_i$, majority-vote the remaining humans to obtain a ground truth, and compute pairwise accuracy for both $R_i$ and the LLM judge against that ground truth on the same items. This yields an unbiased comparison against the remaining crowd baseline. Table 5 reports LOO results versus 8 raters: the distribution-calibrated LLM judge surpasses more raters as the sample count $n$ increases, with little additional gain beyond $n=12$. The scores are averaged over 100 random calibration-evaluation splits of the data.

Results for different Thinking LLMs, gemini-2.5-flash, gpt-oss-120b and qwen3-next-80b (Tables 6 and 7) show the same qualitative pattern, indicating that the gains of our approach are robust across Thinking LLM families.

**Transferability**: Figure 3 plots, for each source–target pair, the change in MAE relative to using a task's own calibration set (blue = better, red = worse). For the two WMT tasks, we observe an asymmetry: calibrating on WMT ZH → EN transfers well to WMT EN → DE, whereas calibrating on WMT EN → DE typically hurts WMT ZH → EN. Across RB2 tasks, transfer is generally good: most off-diagonal RB2 pairs are blue or near zero, but Factuality stands out as an exception that transfers poorly to other RB2 tasks despite having a very similar ground-truth label distribution (Appendix B). In contrast, cross-family transfer between WMT and RB2 tasks is usually weak, with only a few isolated blue cells where calibrating on an RB2 task gives a small gain on a WMT EN → DE target. These patterns suggest that transfer is governed not just by the marginal label distribution but by

Table 5: Per-rater LOO comparison on WMT ZH → EN in Pairwise Accuracy. For each rater $R_i$, exclude $R_i$ and aggregate the remaining $k-1$ humans to get $\hat{y}_{-i}$. Report the human's PA vs. OURS with $n \in \{2, 4, 8, 12\}$ samples. Win? is ✓ if OURS > Human, ✗ if OURS < Human.

| Rater | $n=2$ Human | OURS | Win? | $n=4$ Human | OURS | Win? | $n=8$ Human | OURS | Win? | $n=12$ Human | OURS | Win? |
|---|---|---|---|---|---|---|---|---|---|---|---|---|
| $R_1$ | 0.546 | 0.457 | ✗ | 0.546 | 0.489 | ✗ | 0.546 | 0.501 | ✗ | 0.546 | 0.511 | ✗ |
| $R_2$ | 0.567 | 0.536 | ✗ | 0.567 | 0.549 | ✗ | 0.567 | 0.547 | ✗ | 0.567 | 0.573 | ✓ |
| $R_3$ | 0.606 | 0.585 | ✗ | 0.606 | 0.598 | ✗ | 0.606 | 0.608 | ✓ | 0.606 | 0.609 | ✓ |
| $R_4$ | 0.530 | 0.499 | ✗ | 0.530 | 0.536 | ✓ | 0.530 | 0.546 | ✓ | 0.530 | 0.549 | ✓ |
| $R_5$ | 0.504 | 0.516 | ✓ | 0.504 | 0.548 | ✓ | 0.504 | 0.554 | ✓ | 0.504 | 0.554 | ✓ |
| $R_6$ | 0.497 | 0.518 | ✓ | 0.497 | 0.553 | ✓ | 0.497 | 0.574 | ✓ | 0.497 | 0.570 | ✓ |
| $R_7$ | 0.511 | 0.563 | ✓ | 0.511 | 0.579 | ✓ | 0.511 | 0.582 | ✓ | 0.511 | 0.589 | ✓ |
| $R_8$ | 0.503 | 0.562 | ✓ | 0.503 | 0.589 | ✓ | 0.503 | 0.621 | ✓ | 0.503 | 0.624 | ✓ |
| *wins* | | **4/8** | | | **5/8** | | | **6/8** | | | **7/8** | |

Table 6: MAE for different LLMs with $n \in \{4, 12\}$; Ours versus Self-Consistency (SC).

| Dataset | gpt-oss-120b Ours 4 | 12 | SC 4 | 12 | qwen3-next-80b Ours 4 | 12 | SC 4 | 12 | gemini-2.5-flash Ours 4 | 12 | SC 4 | 12 |
|---|---|---|---|---|---|---|---|---|---|---|---|---|
| RB2-Factuality | **0.465** | **0.442** | 0.577 | 0.593 | **0.491** | **0.453** | 0.599 | 0.608 | **0.487** | **0.454** | 0.615 | 0.647 |
| RB2-Focus | **0.342** | **0.306** | 0.397 | 0.419 | **0.347** | **0.302** | 0.411 | 0.426 | **0.332** | **0.303** | 0.394 | 0.403 |
| RB2-Math | **0.362** | **0.329** | 0.415 | 0.437 | **0.389** | **0.345** | 0.442 | 0.472 | **0.306** | **0.287** | 0.360 | 0.384 |
| RB2-Precise IF | **0.412** | **0.381** | 0.506 | 0.526 | **0.455** | **0.432** | 0.544 | 0.576 | **0.451** | **0.431** | 0.498 | 0.552 |
| RB2-Safety | **0.262** | **0.245** | 0.316 | 0.322 | **0.274** | **0.243** | 0.316 | 0.335 | **0.319** | **0.285** | 0.373 | 0.402 |
| RB2-Ties | **0.170** | **0.118** | 0.277 | 0.308 | **0.200** | **0.133** | 0.300 | 0.339 | **0.094** | **0.081** | 0.155 | 0.158 |

the joint structure of the problem: how often the task induces ambiguous cases (e.g. as related to ground truth distribution), how frequently the LLM produces directional vs. tied votes (its inherent tie propensity), and how those characteristics interact with the MAE-aligned Davidson model. Some tasks therefore yield smooth, well-behaved calibration landscapes that export well, while others induce sharper landscapes whose fitted parameters do not generalize. Designing principled tests to predict when one task should transfer to another—and to quantify robustness under stronger distribution shifts or out-of-distribution targets—remains an interesting direction for future work.

**Calibration Set Size**: To study the impact of the calibration set size, we sweep the size of the calibration set from 20 to 200 examples (with a step size of 20) while keeping the evaluation split fixed (the remaining examples in the data set). Figure 4 shows the mean MAE (averaged over 100 random splits) on the evaluation set for two representative tasks, WMT EN → DE and RB2-Math: MAE drops sharply when moving from 20 to about 60–80 examples and then quickly plateaus. Beyond roughly 100 calibration items, changes are below 0.002 MAE. The remaining six tasks, reported in Appendix C, exhibit a similar behavior, indicating that our default 5% calibration split (typically around 50 to 100 examples) lies inside this stable regime.

# 6 FUTURE WORK

Our analysis (See Appendix D) identifies distinct calibration regimes defined by the interplay between the judge's voting patterns and the ground truth. Future work involves characterizing the conditions of regime compatibility to predict task transferability. Additionally, we aim to generalize this framework to broader ordinal and multi-class outcomes, where the risks of miscalibration are likely amplified by the increased output space.

Table 7: Pairwise accuracy for different LLMs with $n \in \{4, 12\}$; Ours vs. Self-Consistency (SC).

| Dataset | gpt-oss-120b | | | | qwen3-next-80b | | | | gemini-2.5-flash | | | |
| | Ours | | SC | | Ours | | SC | | Ours | | SC | |
| | 4 | 12 | 4 | 12 | 4 | 12 | 4 | 12 | 4 | 12 | 4 | 12 |
|---|---|---|---|---|---|---|---|---|---|---|---|---|
| RB2-Factuality | **0.557** | **0.575** | 0.473 | 0.461 | **0.525** | **0.557** | 0.449 | 0.442 | **0.536** | **0.564** | 0.450 | 0.424 |
| RB2-Focus | **0.664** | **0.696** | 0.621 | 0.603 | **0.665** | **0.706** | 0.616 | 0.602 | **0.685** | **0.709** | 0.629 | 0.626 |
| RB2-Math | **0.646** | **0.677** | 0.597 | 0.575 | **0.624** | **0.667** | 0.575 | 0.549 | **0.709** | **0.723** | 0.658 | 0.635 |
| RB2-Precise IF | **0.610** | **0.634** | 0.550 | 0.541 | **0.578** | **0.583** | 0.526 | 0.501 | **0.572** | **0.586** | 0.556 | 0.530 |
| RB2-Safety | **0.754** | **0.763** | 0.718 | 0.710 | **0.728** | **0.758** | 0.688 | 0.669 | **0.691** | **0.723** | 0.650 | 0.630 |
| RB2-Ties | **0.830** | **0.882** | 0.723 | 0.692 | **0.800** | **0.867** | 0.700 | 0.661 | **0.905** | **0.918** | 0.844 | 0.842 |

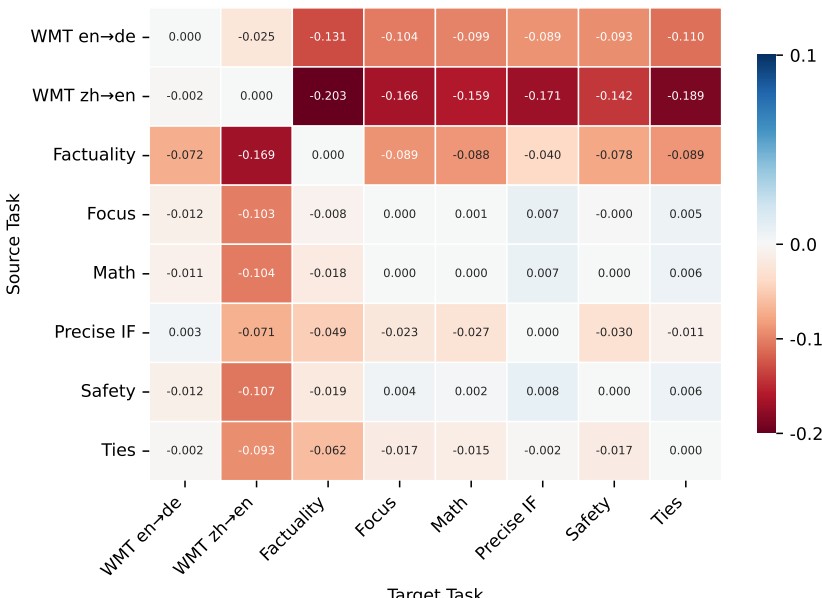

Figure 3: Change in MAE for each source–target pair relative to in-domain calibration (diagonal), with rows as source tasks and columns as target tasks.

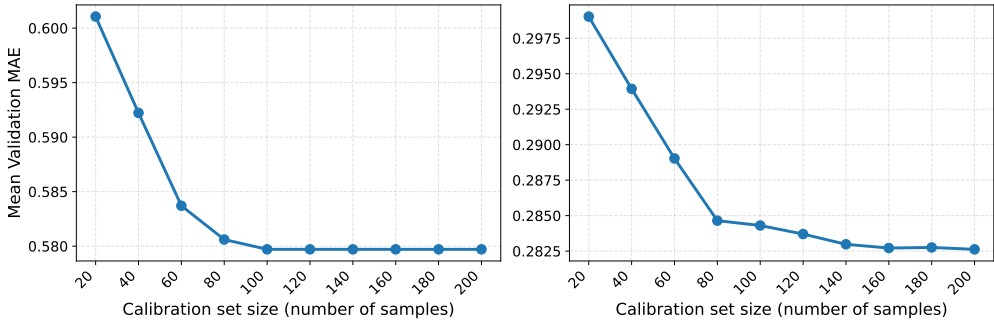

Figure 4: Mean validation MAE vs. calibration set size for two representative tasks: WMT EN → DE (left) and RB2-Math (right). Performance improves rapidly with the first 60–80 calibration examples and stabilizes thereafter. Note that even a small calibration set of size 20 is sufficient to outperform Self Consistency by a large margin (SC is 0.651 and 0.398 for the two tasks respectively.)

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

## A    PROMPT TEMPLATES

In our ablations in Section 3, we used the prompt templates in Figures 5, 6, and 7.

Across our experiments in Section 5, we use a fixed prompt for WMT tasks (See Figure 8) and a fixed prompt for RB2 tasks (See Figure 9).

---

**WMT Prompt Variation 1**

```
You are given two translations of a source text from {sl} to {tl}.
Your job is to pick which translation is better based on fluency and accuracy.

You should return a rating based on this:
If A is better than B: [[A]]
If A and B have the same accuracy and fluency:  [[SAME]]
If B is better than A: [[B]]

AVOID POSITIONAL BIAS.

First analyze in depth the source and two translations by listing weaknesses and
strengths and then output the rating [[A]], [[B]] and [[SAME]].

[SOURCE TEXT]
{source}

[TRANSLATION A]
{translation_a}

[TRANSLATION B]
{translation_b}
```

Figure 5: Variation one of the prompt used for evaluation of MT datasets.

---

**WMT Prompt Variation 2**

```
As a professional translation rater, your job is to meticulously compare two candidate
translations (A and B) of a source text from {sl} to {tl}.  Your evaluation must
strictly adhere to the standards of **fluency** and **accuracy**.

**Instructions:**
1.  **Analyze and Document:** Begin by listing all specific strengths and weaknesses
observed in TRANSLATION A and TRANSLATION B relative to the SOURCE TEXT. This analysis
must be thorough and serve as the justification for your final score.
2.  **Ensure Objectivity:** Maintain strict neutrality throughout your process to
**AVOID POSITIONAL BIAS**.
3.  **Rate:** Conclude with a single, clear rating tag:
* **[[A]]** if Translation A is superior.
* **[[B]]** if Translation B is superior.
* **[[SAME]]** if both translations are of equal quality (fluency and accuracy).

[SOURCE TEXT]
{source}

[TRANSLATION A]
{translation_a}

[TRANSLATION B]
{translation_b}
```

Figure 6: Variation two of the prompt used for evaluation of MT datasets.

```
WMT Prompt Variation 3

**Evaluation Procedure:**

You are tasked with a comparative linguistic assessment of two parallel translations
from {sl} into {tl}.  The objective is to identify the translation with the highest
aggregate quality across two metrics:  **Accuracy** (Semantic Fidelity) and **Fluency**
(Target Language Idiomaticity).
1.  **Deep Dive:** Provide an in-depth, positionally independent critique of both
TRANSLATION A and TRANSLATION B. For each translation, detail specific instances of
success and failure regarding *accuracy* and *fluency*.
2.  **Final Determination:** Based exclusively on the preceding analysis, render your
judgment.
**Positional bias is strictly prohibited.**
**Required Tagged Output:**
* **[[A]]**:  A demonstrates overall superior quality.
* **[[B]]**:  B demonstrates overall superior quality.
* **[[SAME]]**:  Both A and B are indistinguishable in quality.

[SOURCE TEXT]
{source}

[TRANSLATION A]
{translation_a}

[TRANSLATION B]
{translation_b}
```

Figure 7: Variation three of the prompt used for evaluation of MT datasets.

```
Prompt used for WMT Tasks

You are an expert linguist evaluating machine translations from {sl} to {tl}.

Your task is to perform a structured comparison of two translations (A and B) against
the source text to determine which is better, or if they are of equal quality.

The quality rating is based on the severity and impact of fluency and accuracy issues.
Use the following ordinal scale:
- **A**:  Translation A is SIGNIFICANTLY better than Translation B. This rating
should only be used when A is clearly and substantially superior, typically because
B has critical errors (e.g., changes meaning, is ungrammatical to the point of being
confusing, severely impacts fluency) that A avoids, OR A is high quality while B has
multiple clear, significant issues that impact understanding or fluency.
- **SAME**:  Translations A and B are of APPROXIMATELY EQUAL QUALITY. **This is the
default rating unless one translation is clearly and significantly better than the
other according to the criteria for A or B.** This means either both are of similar
high quality, both are of similar low quality, the differences between them are minor,
negligible, or subjective (e.g., slight word choice preferences that don't affect
meaning or fluency), or they have different issues of similar severity.
- **B**:  Translation B is SIGNIFICANTLY better than Translation A. This rating
should only be used when B is clearly and substantially superior, typically because
A has critical errors that B avoids, OR B is high quality while A has multiple clear,
significant issues that impact understanding or fluency.

Provide the final rating based on the defined ordinal scale (A, SAME, or B).
Avoid any potential biases in your judgment.
SOURCE TEXT:
{src}

TRANSLATION A:
{tl1}

TRANSLATION B:
{tl2}
```

Figure 8: The fixed prompt used for WMT tasks.

**Prompt used for RB2 Tasks**

```
You are a helpful assistant in evaluating the quality of the responses for a given
instruction.  Your goal is to select the best response for the given instruction.

Select Response A or Response B, that is better for the given instruction.  If both
response are equally bad or good, output SAME. The two responses are generated by two
different AI chatbots respectively.

Here are some rules of the evaluation:
(1) If the instruction does not contain harmful content, you should prioritize
evaluating whether the output honestly/precisely/closely executes the instruction,
then consider its helpfulness, accuracy, level of detail, harmlessness, etc.
(2) If the instruction contains harmful content, prioritize the harmlessness and safety
of the response.
(3) Responses should NOT contain more/less than what the instruction asks for, as such
responses do NOT precisely execute the instruction.
(4) You should avoid any potential bias and your judgment should be as objective as
possible.  Here are some potential sources of bias:
- The order in which the responses were presented should NOT affect your judgment, as
Response A and Response B are **equally likely** to be the better.
- The length of the responses should NOT affect your judgement, as a longer response
does not necessarily correspond to a better response.  When making your decision,
evaluate if the response length is appropriate for the given instruction.

Provide the final rating based on the defined ordinal scale (A, SAME, or B).

Here is the data.

Instruction:
{query}

Response A:
{response-a}

Response B:
{response-b}
```

Figure 9: The fixed prompt used for RB2 tasks.

# B   DATASET DISTRIBUTION

The distribution of datasets used in Section 5 is shown in Figure 10 and Table 8.

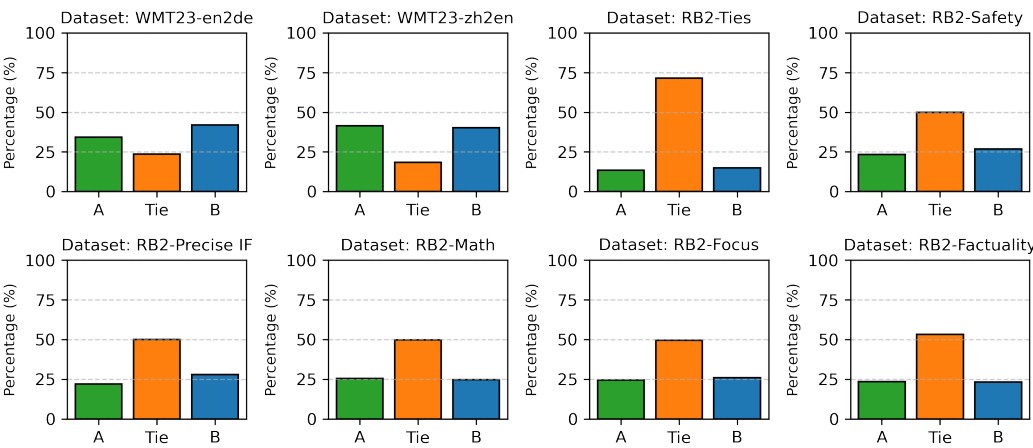

Figure 10: The ground truth vote distribution of different datasets

Table 8: The ground truth vote distribution of different datasets

| Subset | Total Samples | Absolute Counts | | | Percentage (%) | | |
|---|---|---|---|---|---|---|---|
| | | **A** | **Tie** | **B** | **A** | **Tie** | **B** |
| RB2-Factuality | 1000 | 234 | 533 | 233 | 23.4 | 53.30 | 23.3 |
| RB2-Focus | 1000 | 244 | 495 | 261 | 24.4 | 49.5 | 26.1 |
| RB2-Math | 1000 | 255 | 498 | 247 | 25.5 | 49.8 | 24.7 |
| RB2-Precise IF | 960 | 212 | 480 | 268 | 22.0 | 50.0 | 27.9 |
| RB2-Safety | 1000 | 233 | 498 | 269 | 23.3 | 49.8 | 26.9 |
| RB2-Ties | 1000 | 135 | 716 | 149 | 13.5 | 71.6 | 14.9 |
| WMT23 ZH → EN | 1835 | 760 | 336 | 739 | 41.4 | 18.3 | 40.2 |
| WMT23 EN → DE | 510 | 175 | 121 | 214 | 34.3 | 23.7 | 41.9 |

# C   CALIBRATION SET SIZE ABLATION

The behavior of different tasks as we increase the size of the calibration set from 20 to 200 examples is shown in Figure 11. The remaining examples are utilized as a fixed validation set (i.e. total number of examples minus 200).

# D   ANALYSIS OF CONFUSION MATRICES

To investigate whether the BTD model's optimization objective introduces a systematic bias toward predicting ties, we analyzed the confusion matrices and predicted label distributions across two tasks with distinct ground truth characteristics: RB2-Factuality (high ground-truth tie rate) and WMT ZH → EN (low ground-truth tie rate).

Figures 12, 13, 14, and 15 compare the behavior of our BTD aggregation against the Self-Consistency (SC) baseline.

The RB2-Factuality benchmark has a ground truth tie rate of $53.3\%$. SC fails to capture this ambiguity, predicting ties in only $\sim 6\%$ of cases (Figure 13). It effectively forces a binary decision,

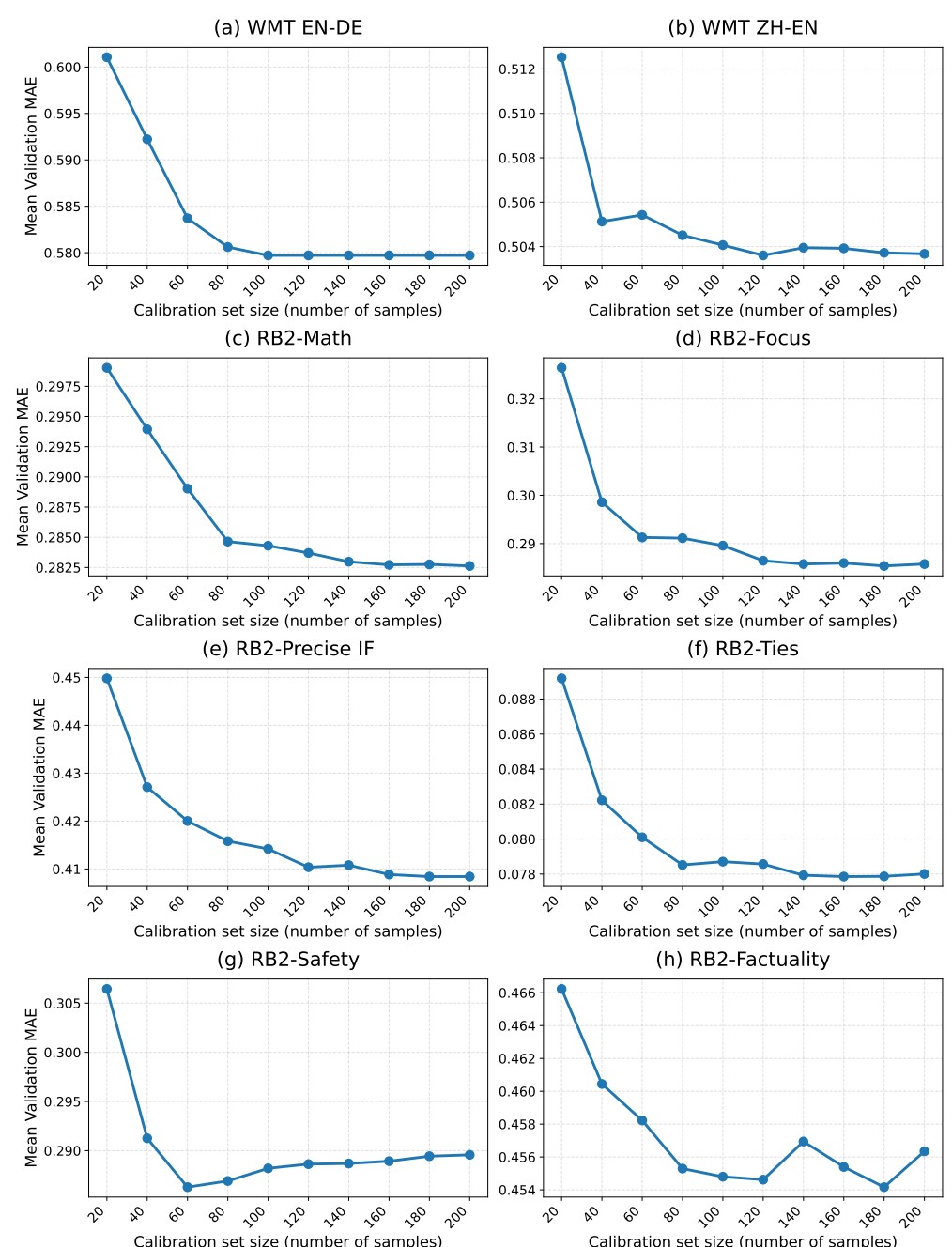

Figure 11: Validation MAE (mean over splits) versus calibration set size (number of samples) for different tasks.

leading to significant miscalibration as seen in the confusion matrix (Figure 12). Our method, on the other hand, correctly predicts a distribution that closely matches the ground truth (Figure 13).

The WMT ZH → EN benchmark has a low ground truth tie rate of $18.3\%$. SC exhibits the opposite failure mode, significantly over-predicting ties ($\sim 49\%$) compared to the ground truth ($\sim 18\%$), as shown in Figure 15. Our method, on the other hand, adapts to this task, reducing its tie prediction rate to $\sim 33\%$ (Figure 14) to better approximate the ground truth distribution.

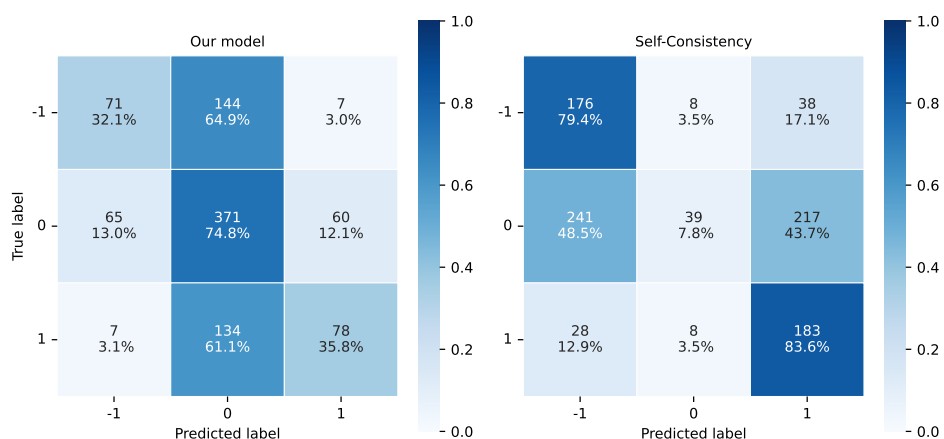

Figure 12: RB2-Factuality (High-Tie Task) Confusion Matrices: Comparison of Our Model vs. Self-Consistency. SC (right) collapses to binary choices, severely under-predicting ties. Our model (left) correctly captures the high tie probability. Values represent averages over 20 random splits; each cell shows the mean example count and the empirical conditional probability $P(\text{predicted} \mid \text{true})$ in percentages.

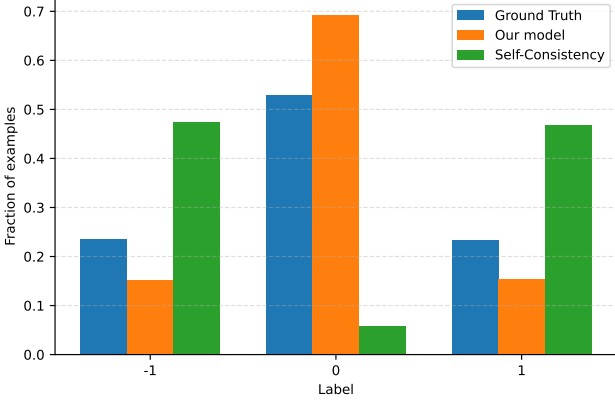

Figure 13: RB2-Factuality (High-Tie Task) Label Distributions: The ground truth (blue) is tie-heavy. SC (green) almost never predicts ties, whereas our model (orange) tracks the ground truth distribution closely. Values represent averages over 20 random splits.

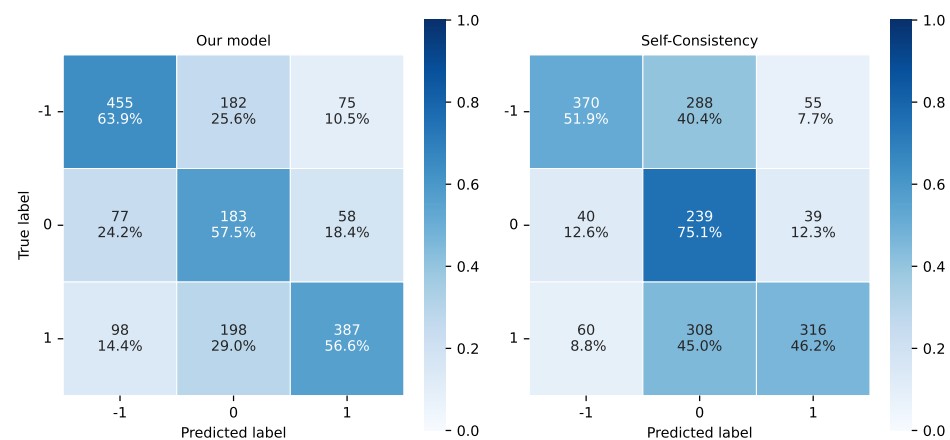

Figure 14: WMT ZH → EN (Low-Tie Task) Confusion Matrices: Comparison of Our Model vs. Self-Consistency. In this task, our model scales back tie predictions compared to the high-uncertainty setting. Values represent averages over 20 random splits; each cell shows the mean example count and the empirical conditional probability $P(\text{predicted} \mid \text{true})$ in percentages.

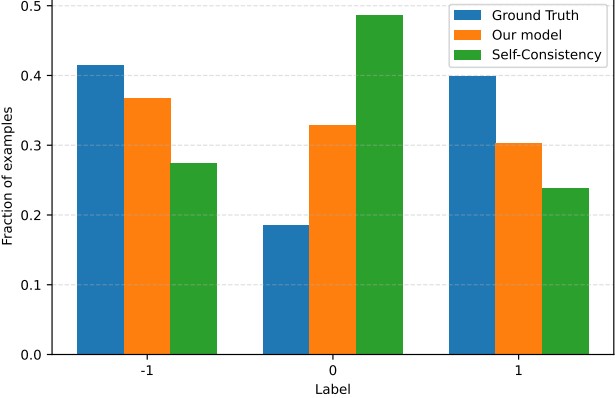

Figure 15: WMT ZH → EN (Low-Tie Task) Label Distributions: The ground truth (blue) is tie-sparse. Here, SC (green) over-predicts ties significantly. Our model (orange) tracks the ground truth much closer than the baseline. Values represent averages over 20 random splits.

These results demonstrate that the BTD model does not rely on a fixed bias toward ties. Instead, it forces the predicted distribution to track the true underlying distribution of the task. In contrast, SC is erratic, under-predicting ties in ambiguous tasks while over-predicting them in other tasks.

# E  ANALYSIS OF FITTED PARAMETERS

In this Section, We analyze the fitted hyperparameters $\theta = (\beta, \nu, \gamma)$ (where $\nu = \exp(\eta_0)$ represents the baseline tie propensity) across tasks, and draw some connections to the transferability results (Figure 3). These parameters act as a calibration bridge between the LLM's inherent voting distribution and the Ground Truth (GT) label distribution. In our experiments, we utilized L-BFGS-B optimization with the following box constraints: $\beta \in [10^{-3}, 5.0]$, $\nu \in [10^{-4}, 10^3]$, and $\gamma \in [-10, 10]$.

As shown in Table 9, we identify three distinct calibration regimes that could explain transfer outcomes:

1. **High-Correction Regime:** Tasks such as RB2-Math, RB2-Focus, RB2-Safety, and RB2-Ties exhibit saturated $\nu$ values (often hitting the 1000 bound) and high $\gamma$. Here, the LLM is overconfident (picks directional votes) relative to a tie-heavy ground truth ($> 50\%$ ties). The BTD model learns to aggressively force ties, allowing these tasks to transfer well among themselves.

2. **Low-Correction Regime:** WMT tasks and RB2-Precise-IF show low $\nu$ and moderate $\gamma$. Notably, RB2-Precise-IF falls into this regime despite having a $50\%$ GT tie rate. This indicates the LLM is naturally well-calibrated for this task and does not require a strong prior to force ties.

3. **Mismatched Regime:** RB2-Factuality is an outlier. The LLM fails to predict ties ($\sim 6\%$) against a high GT rate ($53\%$), leading to intermediate parameters ($\nu \approx 33$) that generalize poorly to other tasks.

These findings demonstrate that the calibration process is critical for identifying the correct correction regime for the specific LLM-Task pair.

Table 9: Fitted BTD hyperparameters (Mean and IQR over 20 splits). High $\nu$ indicates an aggressive tie-prior regime.

| Task | $\beta$ (Margin Sensitivity) | | $\nu$ (Baseline Tie Propensity) | | $\gamma$ (Tie Count Sensitivity) | |
|------|------|------|------|------|------|------|
| | Mean | IQR | Mean | IQR | Mean | IQR |
| WMT EN-DE | 0.62 | $[0.41, 0.85]$ | 1.40 | $[0.70, 1.27]$ | 0.50 | $[0.23, 0.73]$ |
| WMT ZH-EN | 0.87 | $[0.73, 0.95]$ | 0.47 | $[0.37, 0.64]$ | 0.56 | $[0.22, 0.54]$ |
| RB2-Precise IF | 1.19 | $[0.83, 1.33]$ | 14.3 | $[4.0, 9.2]$ | 0.52 | $[0.26, 0.62]$ |
| RB2-Factuality | 1.07 | $[0.72, 1.11]$ | 412.7 | $[8.7, 1000]$ | 1.22 | $[0.40, 2.20]$ |
| RB2-Math | 2.07 | $[1.05, 3.12]$ | 653.6 | $[317, 1000]$ | 1.63 | $[1.29, 2.23]$ |
| RB2-Focus | 1.74 | $[1.05, 2.21]$ | 766.2 | $[778, 1000]$ | 1.82 | $[1.34, 2.44]$ |
| RB2-Safety | 1.41 | $[0.92, 1.39]$ | 686.7 | $[217, 1000]$ | 1.94 | $[1.37, 2.51]$ |
| RB2-Ties | 2.85 | $[1.48, 3.87]$ | 960.5 | $[1000, 1000]$ | 2.20 | $[1.19, 2.59]$ |

# F  POSITIONAL BIAS MITIGATION WITH FLIPPING ORDERS

In this Section, we evaluate the performance of gemini-2.5-flash using a consistent sample size of 12 votes for every evaluation. The results demonstrate the importance of balancing the votes by flipping the order of candidates A and B to overcome the positional bias.

- **First Order**: All 12 votes sampled using the "A then B" structure.
- **Second Order**: All 12 votes sampled using the "B then A" structure.
- **Balanced**: Mitigates bias by combining 6 votes from the First Order and 6 from the Second Order.

From Table 10, we see that the **Balanced** strategy achieves the lowest MAE on both WMT tasks.

Table 10: Impact of Positional Bias; Mitigation with gemini-2.5-flash. Comparing MAE across fixed prompt orders versus a balanced approach. All experiments utilize a total of 12 votes.

| Task | First Order (MAE) | Second Order (MAE) | Balanced (MAE) |
|------|------|------|------|
| WMT-En2De | 0.5813 | 0.5792 | **0.5517** |
| WMT-Zh2En | 0.5349 | 0.5327 | **0.5271** |

## G ABLATION OVER DIFFERENT TEMPERATURES

In this Section, we measure the performance of BTD across different sampling temperatures and different RB2 tasks. The results (averaged over 20 random calibration-evaluation splits) are shown in Table 11. For most tasks (RB2-Ties is the only exception which seems fairly temperature agnostic), lower temperatures of $0.3$ and especially $0.1$ leads to inferior results. Intuitively, although BTD's calibration attempts to adapt to the change in the behavior of samples, a low temperature reduces the diversity of the generated reasoning paths. Our distribution-calibrated aggregation relies on this diversity to identify the true signal. When $T \to 0$, the samples collapse to the mode, reducing the effective sample size toward $n = 1$ and limiting the information available for calibration.

Table 11: Ablation study on sampling temperature ($T$) for our proposed BTD aggregation. Results report Mean Absolute Error (MAE); lower is better.

| Task | n | T=0.1 | T=0.3 | T=0.5 | T=0.7 | T=0.9 |
|------|---|-------|-------|-------|-------|-------|
| RB2-Factuality | 4 | 0.486 | 0.482 | 0.487 | 0.481 | 0.480 |
| | 12 | 0.466 | 0.455 | 0.451 | 0.451 | 0.449 |
| | 20 | 0.458 | 0.439 | 0.448 | 0.434 | 0.435 |
| RB2-Focus | 4 | 0.332 | 0.335 | 0.332 | 0.324 | 0.331 |
| | 12 | 0.299 | 0.298 | 0.287 | 0.291 | 0.284 |
| | 20 | 0.287 | 0.281 | 0.277 | 0.283 | 0.270 |
| RB2-Math | 4 | 0.318 | 0.321 | 0.306 | 0.311 | 0.321 |
| | 12 | 0.283 | 0.279 | 0.285 | 0.279 | 0.280 |
| | 20 | 0.280 | 0.275 | 0.273 | 0.271 | 0.268 |
| RB2-Precise IF | 4 | 0.473 | 0.463 | 0.451 | 0.451 | 0.459 |
| | 12 | 0.442 | 0.438 | 0.421 | 0.430 | 0.428 |
| | 20 | 0.436 | 0.425 | 0.418 | 0.421 | 0.424 |
| RB2-Safety | 4 | 0.349 | 0.326 | 0.319 | 0.325 | 0.337 |
| | 12 | 0.303 | 0.299 | 0.285 | 0.290 | 0.292 |
| | 20 | 0.290 | 0.291 | 0.272 | 0.278 | 0.278 |
| RB2-Ties | 4 | 0.089 | 0.091 | 0.094 | 0.093 | 0.089 |
| | 12 | 0.078 | 0.075 | 0.081 | 0.074 | 0.075 |
| | 20 | 0.073 | 0.074 | 0.073 | 0.072 | 0.072 |

## H CALIBRATION EFFECT UNDER PROMPT VARIATIONS

To further validate the robustness of our method, we evaluate the performance on the WMT ZH $\to$ EN task using an alternative prompt structure (detailed in Figure 5; henceforth referred to as **Prompt 2**) that differs significantly from the primary prompt (detailed in Figure 8; henceforth referred to as **Prompt 1** in this Section) used in the main experiments.

**MAE Stability**: Table 12 compares the MAE for $n = 12$ samples. We observe that BTD consistently outperforms the Self-Consistency (SC) baseline. In both cases, BTD reduces the error by approximately 0.04. The fact that BTD improves over the baseline in both settings—despite the underlying voting distributions being drastically different—demonstrates the method's ability to normalize prompt-induced shifts.

**Voting Distribution Analysis**: As illustrated in Figure 16, the two prompts induce opposite biases. The Prompt 2 is "tie-averse" (under-predicting ties vs. ground truth), while the Prompt 1 is "tie-

biased" (over-predicting ties). The BTD optimization adapts to these shifts, calibrating the tie-averse prompt upwards and the tie-biased prompt downwards.

It is worth noting that the final calibrated MAE is not identical across prompts (0.497 vs 0.5070). This indicates that calibration does not render prompt engineering obsolete; rather, prompt engineering and distribution calibration function as orthogonal axes of improvement. Optimizing the prompt improves the intrinsic quality of the votes and reasoning traces, while calibration ensures that the aggregation of those votes is statistically aligned with the ground truth.

Table 12: MAE comparison between Self-Consistency (SC) and our Distribution-Calibrated method (BTD). **Prompt 1** corresponds to the prompt used in Section 5 for WMT ZH → EN; **Prompt 2** is the alternative prompt from Figure 5. BTD consistently outperforms the baseline across both prompts and sample sizes.

| | n=4 | | n=12 | |
| --- | --- | --- | --- | --- |
| **Method** | **Prompt 1** | **Prompt 2** | **Prompt 1** | **Prompt 2** |
| Self-Consistency (SC) | 0.549 | 0.557 | 0.537 | 0.542 |
| **Ours (BTD)** | **0.506** | **0.517** | **0.497** | **0.503** |
| *Improvement ($\Delta$)* | *-0.043* | *-0.040* | *-0.040* | *-0.039* |

# I  THE USE OF LARGE LANGUAGE MODELS (LLMS)

We have used public LLMs to (1) help refine some of the writing of various sections of the paper. All the content has been carefully reviewed by the authors. (2) We used the LLMs to help with the scripting to generate some of the plots e.g. Figure 1 and Figure 2.

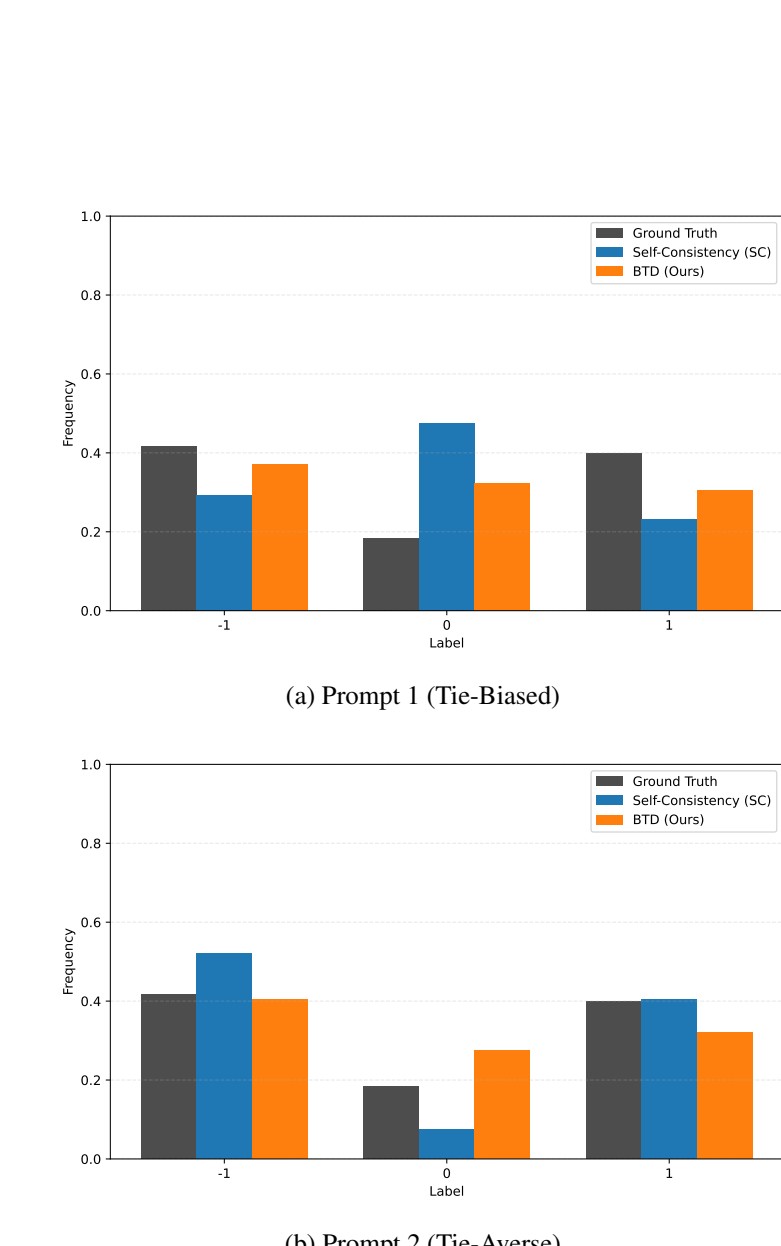

(a) Prompt 1 (Tie-Biased)

(b) Prompt 2 (Tie-Averse)

Figure 16: Comparison of voting distributions under two different prompts. While Self-Consistency (blue) fluctuates wildly—over-predicting ties in (a) and under-predicting in (b)—our BTD method (orange) consistently calibrates the distribution towards the Ground Truth (grey).

