# OpenReview forum: "Distribution-Calibrated Inference Time Compute for Thinking LLM-as-a-Judge"
_ICLR.cc/2026/Conference — ICLR 2026 Conference Desk Rejected Submission_

### Official Review · Reviewer_1jJ9 · 2025-10-18

**Soundness:** 3
**Presentation:** 4
**Contribution:** 4
**Rating:** 8
**Confidence:** 4

**Summary:**

This paper introduces a novel distribution-calibrated aggregation strategy for LLM-as-a-Judge in pairwise preference tasks. The authors first highlight the importance of allowing "tie" judgments to mitigate LLM biases and then demonstrate that tie-rates are highly sensitive to prompt and model variations. The core proposal is a principled aggregation scheme based on a Bradley–Terry-Davidson model that leverages the full distribution of votes from multiple inference-time samples. By considering both the margin between preferences and the rate of non-ties, the method aims to create more reliable and robust evaluations that align better with human judgment.

**Strengths:**

- The paper is well-motivated and effectively presented. It clearly articulates the problem with existing aggregation methods, particularly their failure to handle ties and distributional information gracefully. The manuscript, while concise, is complete and easy to follow.

- The proposed method is well-grounded in established statistical principles, using a Bradley–Terry-Davidson formulation to model three-way preferences. This provides a strong theoretical foundation for the approach.

- The theoretical claims are supported by a thorough empirical evaluation across several benchmarks. The consistent improvements in MAE and pairwise accuracy over standard baselines effectively demonstrate the value of the proposed aggregation scheme.

**Weaknesses:**

See questions.

**Questions:**

- In Section 3, the authors convincingly demonstrate that the rate of "tie" judgments is highly sensitive to minor variations in prompt wording. However, it is unclear if the main experiments in Section 5 were conducted using a single, fixed prompt. To fully validate the robustness of the proposed distribution-calibrated method, it would be beneficial to show its performance across the different prompt variations introduced earlier. This would demonstrate that the calibration process can successfully normalize for prompt-induced shifts in the voting distribution.

- The proposed method relies on a calibration set to fit the parameters. The size of this set is a critical hyperparameter that likely affects the final performance. The paper currently uses a fixed percentage of the data for calibration but does not explore how performance changes with varying sizes. It is recommended to conduct an ablation study on the size of the calibration set. This would provide valuable insight into the method's data efficiency and offer practical guidance for users on how large a calibration set is required to achieve stable and reliable results.

---

> ### Author Response · Authors · 2025-11-26
>
> Q (Prompt Choice; Prompt Variation Effect):
>
> The experiments in Section 5 were carried out using a fixed single prompt: one fixed prompt for the WMT tasks, and one fixed prompt for the RB2 tasks. Please see Appendix I for the prompts.
>
> To analyze the prompt variation effect, we add Appendix H. In this Appendix, we use an alternative prompt for the WMT zh->en task, and compare the results against those of Section 5 with the primary prompt. Note that the two prompts are significantly different, leading to votes with very different behavior. One prompt triggers a tie-averse behavior (LLM rarely produces ties: ) whereas the other is tie-biased (LLM produces a lot of ties).
>
> 1. Consistent Error Reduction: As shown in Table 12, our method outperforms the standard Self-Consistency (SC) baseline in MAE across both prompts. Crucially, the improvement gap remains stable ($\Delta \approx 0.04$) regardless of the prompt used. This confirms that our calibration method reliably extracts better signal from the noisy vote distribution, regardless of the specific phrasing of the prompt.
>
> 2. Mechanism of Stability: We analyzed the voting histograms (see Figure 16 in the updated manuscript) to understand this stability.
>
> Prompt 2 resulted in a "tie-averse" distribution (predicting ties only 8% of the time). BTD successfully calibrated this upwards to match the ground truth (19%).
>
> Prompt 1 resulted in a "tie-biased" distribution (predicting ties 48% of the time). BTD successfully calibrated this downwards.
>
> Q: (Size of the Calibration set):
>
> We agree that understanding the dependence on calibration set size is important.
> In the revised manuscript, we have added an ablation that sweeps the calibration size from 20 to 200 items (with a step size of 20). Figure 4  in the main paper shows results for two representative tasks (WMT en->de and RB2-Math), and Appendix C reports the curves for all eight tasks.
> Across datasets, MAE decreases rapidly when increasing the calibration set from 20 to about 60–80 examples and then plateaus; beyond roughly 100 calibration items, additional gains are below 0.002 MAE. This indicates that our default 5% calibration split (typically around 50-100 examples) is already in a stable regime and that the method is data-efficient in practice.

---

> > ### Comment · Reviewer_1jJ9 · 2025-11-27
> >
> > Thanks for author response which resolved my concerns. I believe the current score is a fair assessment of the paper, and I will retain it.

---

### Official Review · Reviewer_FtNF · 2025-10-26

**Soundness:** 3
**Presentation:** 3
**Contribution:** 2
**Rating:** 4
**Confidence:** 4

**Summary:**

This paper proposes a Distribution-Calibrated Aggregation method to improve the aggregation of thinking LLM-as-a-Judge. They first address two interesting observations that motivate their research: (1) ties are important to reduce LLM biases, and (2) Tie decisions are not stable. Based on the observations, they draw a conclusion that tie is important for LLM-as-a-judge and aggregation is needed as tie is not stable. Then, they leverage the Bradley-Terry-Davidson framework to estimate aggregation parameters based on a calibrated subset. They perform empirical studies across MT and Reward Bench tasks to show that their method outperforms baselines.
Overall, this paper could be a nice contribution to LLM-as-a-Judge research, with the clarifications on the importance of tie, a parameter fitting method based on BT, and experiments across datasets. I have some concerns about the experiments and analysis (see weakness), but I would be willing to increase the score with the author's clarification.

**Strengths:**

1. Clear research motivation addressing the importance of tie decisions in reducing LLM biases and their instability for LLM-as-a-Judge.
2. The method is theoretically rigorous and easy to understand.
3. Sufficient and broad experimental validation across MT and Reward Bench tasks.

**Weaknesses:**

1. In comparison to the baselines, the paper uses an additional calibration set (5%-10% of test data). This indicates additional human effort when using the method for different tasks. Is it possible to fit BTD on one task and test on the other task? Or will the parameter of BTD be very sensitive to the selected task?
2. Based on argument 1, a sensitive analysis should be presented if the proposed method is not fitted with the best parameters.
3. Is thinking important? The proposed method is applied to the Thinking LLM-as-a-Judge; however, a general LLM-as-a-Judge may encounter the same issue. Is this because the Thinking LLM-as-a-Judge typically outperforms the general LLM-as-a-Judge? Or does the Thinking LLM-as-a-Judge introduce greater bias, thereby creating a stronger need for the proposed aggregation method?
4. To address the second motivation, prompt template 3 naturally leads to different results as it gives the definition of accuracy and fluency, e.g.,  Accuracy can refer to both Semantic Fidelity and Lexical Fidelity.
5. The proposed method lacks analysis with different variants, e.g., what if the model is not optimized based on accuracy instead of MAE?

Things to improve the paper that did not impact the score:
1. Lack of citations about pairwise LLM-as-a-Judge and its bias:
[1] Liu Z, Wang P, Xu R, et al. Inference-time scaling for generalist reward modeling[J]. arXiv preprint arXiv:2504.02495, 2025.
[2] Ye Z, Li X, Li Q, et al. Learning LLM-as-a-judge for preference alignment[C]//The Thirteenth International Conference on Learning Representations. 2025.
[3] Junsoo Park, Seungyeon Jwa, Meiying Ren, Daeyoung Kim, and Sanghyuk Choi. Offsetbias: Leveraging debiased data for tuning evaluators. arXiv preprint arXiv:2407.06551, 2024.
[4] Zheng C, Zhou H, Meng F, et al. Large language models are not robust multiple-choice selectors[J]. arXiv preprint arXiv:2309.03882, 2023.
2. Presentation quality, e.g., lack of colon in line 247, strange italic in line 245; the algorithm is difficult to read without the context.

**Questions:**

1. Note that stratification of the splits empirically did not change the results. -> Not clear.
2. According to Figure 7, ties are very common for BTD. Will BTD tend to tie based on its optimization target? Could you provide a confusion matrix of BTD and compare it with other methods?
3. Why do you use different models in Table 1 and Table 2?
4. What's the selection of the fitting parameters in different datasets, and what can we observe from these selections?
5. Will the temperature affect the results?

---

> ### Author Response · Authors · 2025-11-26
>
> Q1: (Additional Citations):
>
> We thank the reviewer for pointing out these relevant references. We have incorporated all of them into the revised manuscript since they provide valuable context to our work. We particularly cite [2] and [3] as methods used to debias models through training techniques, [1] as an inference time method used for reward modeling and [4] as another source to showcase the instability of LLMs as a judge.
>
> Q2: (Stratification Clarification):
>
> We removed the confusing sentence about stratification. To clarify, we found that stratifying the calibration split (i.e. ensuring class balance so that the calibration set has the same distribution of class A, class ties, and class B as the eval set), yielded no significant performance difference compared to random sampling.
>
> Q3: (Bias towards ties):
>
> We appreciate this crucial question. We investigated whether the BTD objective creates a systematic bias toward ties. Our analysis in Appendix D confirms that it does not; instead, BTD adapts to the true underlying distribution of the task, whereas the self-consistency (SC) baseline drifts erratically.
> To demonstrate this, we investigated the confusion matrices and predicted label distributions for two tasks with opposite characteristics:
> - RB2-Factuality (High Tie Regime ~53%): As shown in Figure 8, the ground truth is dominated by ties. Self-Consistency (SC) fails here by under-predicting ties (predicting only ~6%), effectively forcing incorrect binary choices. Our BTD model correctly captures the high ambiguity, matching the ground truth distribution.
> - WMT ZH->EN (Low Tie Regime $\textasciitilde$ 18%): In this task (new Figure 9), the signal is clearer. Interestingly, SC fails here by over-predicting ties (~49%), likely due to the fragmented voting distribution. In contrast, BTD adapts, reducing its tie rate significantly to track the lower ground truth rate.
> In conclusion, BTD acts as a calibration mechanism that anchors the model to the true distribution of labels, rather than blindly favoring ties.
>
> Q4: (Model Selection in Tables 1 & 2):
>
> We tested all models but prioritized those showing significant bias for the exposition. In table 1, both deepseek v3.1 and gpt-oss-120b had less than 1% positional bias. We updated the draft to point this out (line 140:  “Both other models, gpt-oss-120b and deepseek-v3.1, had less than 1% positional bias in this setup and thus were not included in the table.”)
>
> Q5: (Calibration effort, Transferability, values of the fitted parameters and their sensitivity):
>
> - Calibration effort: while we use a calibration set, our ablation in Figure 4 shows that performance stabilizes with as few as 40-60 examples. Even 20 examples are sufficient to significantly outperform the self consistency baseline. Labeling 20 examples is a minimal human effort burden compared to the significant gains in reliability.
> - Transferability: We performed the requested cross-task experiments. Please see the transferability subsection added to Section 5 (and Figure 3).
> - Parameter selection and sensitivity: We added a detailed parameter analysis in Appendix E. We found that parameters of our 8 tasks typically settle into certain regimes based on the model’s inherent overconfidence relative to the ground truth distribution of the specific task. The parameter analysis is also closely tied to the transferability across different tasks.
>
> Q6: (Temperature Ablation):
>
> We added an ablation study on sampling temperature in Appendix G. We found that lower temperatures ($T=0.1, 0.3$) yield suboptimal results compared to higher temperatures ($T=0.5, 0.7, 0.9$). This indicates that diversity in the reasoning paths helps our aggregation method.
>
> Q7: (Importance of Thinking)
>
> The "Thinking" process is crucial for our method for two main reasons:
> - Performance: As established in recent work (Saha et al., 2025), Thinking LLMs (using Chain-of-Thought) outperform direct-answer models on complex evaluation tasks. For reference, we compared the performance of gemini-flash under non-thinking mode with its thinking mode under greedy decoding over WMT tasks. In both cases, the thinking mode outperformed the non-thinking mode by a large margin. For zh->en, MAE (GD, thinking mode) = 0.527, MAE (GD, non-thinking mode) = 0.566. For en->de, MAE (GD, thinking mode) = 0.651, MAE (GD, thinking mode) = 0.663. Note that for GD we draw two samples with flipped orders and aggregate them using the rounded median, as described in the draft under the description of baselines.
> - The thinking process generates diverse reasoning traces. Our temperature ablation (mentioned above) empirically validates that diversity in the reasoning traces helps the aggregation method: performance degrades at low temperatures ($T \in \{0.1, 0.3\}$) where reasoning diversity collapses, demonstrating that our distribution-calibrated aggregation relies on exploring this latent reasoning space to function effectively.

---

> > ### Comment · Reviewer_FtNF · 2025-11-27
> > **Official Comment by Reviewer**
> >
> > Thank you for acknowledging and updating the manuscript based on some of the questions. I have increased my score as I believe that this work is a positive improvement for LLM-as-a-Judge.

---

### Official Review · Reviewer_jhV9 · 2025-10-28

**Soundness:** 2
**Presentation:** 2
**Contribution:** 2
**Rating:** 6
**Confidence:** 4

**Summary:**

This paper addresses the inconsistency of existing aggregation methods for LLM-as-a-judge systems by introducing a distribution-calibrated aggregation scheme that models the full distribution of multiple reasoning–rating samples. Using a Bradley–Terry–Davidson formulation, the method explicitly accounts for both preference margins and tie propensity, aligning inference with the MAE evaluation metric through maximum-likelihood calibration. Experiments across translation and reward evaluation benchmarks show that this approach consistently improves pairwise accuracy and matches or exceeds human rater performance.

**Strengths:**

- The paper is good in presentation, though it seems very obvious that the authors are trying very hard to stretch their content to 9 pages.
- It is quite novel to reframe LLM judgement aggregation using a Bradely Terry Model.
- The experiments show improvements over baselines.

**Weaknesses:**

- In the motivation section, the author appears to make a very strong claim that 'Ties are important to reduce LLM biases. ' Though it reduces the score in numbers, it doesn't really solve the fundamental problem of the model computation mechanism that directly related to position bias. As noted by Wang et al.  [1], the core problem is the mechanism of position bias. Because of this, the claim of 'Ties are important to reduce LLM biases' is way too strong. Instead of reducing it, it seems more likely to hide the problem of position bias.
- The author claims that  'In other baselines, in order to overcome the positional bias, we draw n/2 samples in an A-then-B response order and the remaining n/2 samples via a B-then-A order.', however, there is really no experiments nor results to validate the effect of this setup. Does this actually work?
- For table 3, and table 4, it only shows the results from one model. I wonder how other models performed? And only three models are tested unfortunately. How about Claude and deepseek models? And how about Judgebench[2] that is specifically designed for benchmarking llm judges.

[1] Eliminating Position Bias of Language Models: A Mechanistic Approach

[2] JudgeBench: A Benchmark for Evaluating LLM-based Judges

**Questions:**

Please refer to weakness

---

> ### Author Response · Authors · 2025-11-26
>
> Q1: (Bias Claim):
>
> We agree that introducing a tie option does not solve the fundamental computational mechanism of position bias (as noted by Wang et al.). What we are arguing is that in a practical evaluation setting, a forced choice amplifies bias by forcing the model to rely on positional priors when it is uncertain. We will update the draft to point this out and add the citation. Thanks!
>
> Q2: (Effectiveness of Flipping Order):
>
> Addressing positional bias by permuting the order of candidate responses is a standard solution in LLM evaluation (Zheng et al., Judging LLM-as-a-Judge, 2023). As Zheng et al. note, "Position bias can be addressed by... swapping the order of two answers and only declaring a win when an answer is preferred in both orders."
> We added an ablation study (details added to Appendix F) to empirically verify this. We found that combining votes from flipped orders ($n/2$ split) reduced the Mean Absolute Error by approximately 4-5% compared to using a fixed order across both WMT-En2De and WMT-Zh2En tasks, empirically validating the effectiveness of this setup.
>
> | Task            | First Order (MAE) | Second Order (MAE) | Flipping Order (MAE) |
> |-----------------|-------------------|---------------------|----------------------|
> | WMT-En→De       | 0.5813            | 0.5792              | **0.5517**           |
> | WMT-Zh→En       | 0.5349            | 0.5327              | **0.5271**           |
>
>
> Q3: (Model & Benchmark Coverage):
>
> We were limited by compute budget and thus we only limited the extensive results to gemini. We suggest that with three models (gemini, gpt, qwen; note that the results for the latter models are shown in Tables 6 and 7) and 8 different tasks across two separate benchmarks (WMT and RB2), the results provide us with sufficient confidence that the findings will generalize. Please let us know if you think it is critical to expand either model or dataset coverage and we will add them. Thank you!

---

### Official Review · Reviewer_C1Rb · 2025-11-01

**Soundness:** 3
**Presentation:** 3
**Contribution:** 2
**Rating:** 4
**Confidence:** 3

**Summary:**

This paper addresses a critical limitation of Thinking LLM-as-a-Judge—the noise in single-sample pairwise preference judgments and the statistical suboptimality of existing aggregation methods (e.g., majority vote, soft self-consistency) when ties are allowed. The core contribution are:
(1) Demonstrating suboptimality of existing ITC aggregation methods for LLM judges;
(2) Proposing an ERM-based BTD aggregation scheme that outperforms baselines across tasks;
(3) Developing a consensus-based meta-evaluation for noisy labels, enabling fair human-LLM comparisons.

**Strengths:**

1, The paper identifies and addresses two understudied gaps in LLM-as-a-Judge research—tie instability across models/prompts and the loss of evidential strength in simple aggregation. The BTD-based distribution-calibrated scheme is a creative adaptation of statistical preference modeling to LLM evaluation, filling a critical methodological gap.

2,  Experiments are rigorous and comprehensive. Testing across 3 LLMs (gemini-2.5-flash, qwen3-next-80b, gpt-oss-120b) and 8 benchmarks ensures generalizability.

3, The paper balances technical depth with accessibility.

4, Theoretically, it establishes a principled framework for ITC aggregation that aligns with evaluation metrics.

**Weaknesses:**

1, More related works should be discussed. e.g. https://aclanthology.org/2024.findings-emnlp.135.pdf, https://arxiv.org/abs/2401.02009, https://arxiv.org/abs/2308.00436. For example, at the same cost, does the proposed method perform better than mirror-consistency, self-contrast & self-check?

2, Generating  samples and fitting the BTD model adds computational cost. The paper does not quantify this overhead (e.g., inference time, token usage) relative to baselines e.g., Self-Consistency with n=12, making it hard to assess practicality for real-time applications.

3,  The method requires a labeled calibration set. The paper does not explore scenarios where calibration data is scarce (e.g., few-shot/zero-shot settings) or domain-shifted.

**Questions:**

1, What's the performance comparison between the consistency-based methods and other inference-time methods? e.g. multi-agent systems or other prompting methods like step-back https://arxiv.org/abs/2310.06117. Or let me ask in another way, why should we keep optimizing consistency-based methods, given all other prompting strategies?

2, The method is mainly a prompting engineering work. Can the llm be trained to be better at calibration?

3, Have you tested if calibration parameters learned on one task (e.g., RB2-Factuality) can be transferred to another (e.g., WMT ZH→EN)? If so, how much performance is lost compared to task-specific calibration?

---

> ### Author Response · Authors · 2025-11-26
>
> [1] ​​(Related Works):
>
> We thank the reviewer for these excellent references. We added the following discussion of them to the Related Work section:
>
> “Generator Refinement and Verification: A different line of work refines the generation process itself. Methods like Mirror-Consistency, Self-Contrast, and Step-Back Prompting utilize iterative reflection, diverse perspectives, or abstraction to produce higher-quality samples, while Self-Check employs step-wise verification to filter errors. Unlike these approaches, which focus on enhancing the generator (often incurring sequential computational costs), our work focuses on the aggregator: we accept the noisy distribution of parallel samples and apply a principled, distribution-calibrated layer to robustly estimate the ground truth.”
>
> While we cannot run new concurrent experiments with these specific baselines in the rebuttal window, we emphasize three critical distinctions that highlight the advantages of our approach:
>
> 1. Sequential vs. Parallel Inference-Time Compute (Cost & Latency):
> The suggested methods (Mirror-Consistency, Self-Contrast, Self-Check) are primarily sequential and iterative.
> Self-Check [3] requires regenerating individual reasoning steps to verify consistency, leading to a multiplicative increase in inference calls ($N \times \text{steps}$).
> Mirror-Consistency [1] relies on an iterative "reflection" loop where the model must critique previous outputs.
> Self-Contrast [2] mandates generating diverse perspectives followed by a summarization step.
> These methods incur high latency (due to serial dependency) and significantly higher token costs (as context grows with critiques/history). In contrast, our BTD method is a parallel aggregation technique. It operates on standard independent samples, allowing for full parallelization and zero additional generation latency. In real-time applications, this parallel efficiency is often decisive.
>
> 2. Orthogonality (Generator vs. Aggregator):
> We view these methods as orthogonal to ours. Techniques like Self-Contrast or Step-Back aim to improve the quality of the individual sample (the generator). Our method aims to improve the estimation of the ground truth from a noisy distribution (the aggregator).
> As shown in the literature (e.g., Inference-Time Scaling [4]), better sampling strategies can be combined with better aggregation. One could theoretically use Self-Contrast to generate higher-quality samples and then use our BTD model to aggregate them. Therefore, our method complements rather than competes with these prompting strategies.
>
> 3. Robustness to Intrinsic Bias:
> Iterative self-correction methods rely on the model's ability to recognize its own errors. However, models often exhibit "overwhelming bias" where they are confidently wrong.
> For example, in our new results for RB2-Factuality (Appendix D), we observe an extreme directional bias where the model rarely predicts ties, despite the ground truth being tie-heavy.
> In such "high-bias" regimes, self-critique methods often fail because the model's internal prior is too strong to be overturned by its own reflection. Our distribution-calibrated approach does not ask the model to "fix" itself; instead, it learns a lightweight correction layer that maps the model's biased output distribution to the true label distribution, effectively solving the issue where self-correction would likely fail.
>
> [2] (Computational Cost):
>
> The computational overhead of our method is negligible compared to the baseline.
> - Inference Time: The "heavy lifting" is the generation of the $n$ samples by the LLM. Once those samples are generated, the BTD aggregation (calculating features $s$ and $t$ and applying the Bayes decision rule) is a lightweight arithmetic operation that takes microseconds. Therefore, the latency of our method with $n=12$ is effectively identical to Self-Consistency with $n=12$.
> - Token Usage: There is no additional token usage during inference compared to standard Self-Consistency.
> - Calibration Cost: The parameter fitting (minimizing DRPS) is an offline process performed once per task. As shown in our ablation studies (Figure 4), this requires only a small set of samples (~40–60), making the setup cost very low.

---

> ### Author Response · Authors · 2025-11-26
>
> [3] (Scarcity of Calibration Data):
>
> We added new results to the Experiments Section to explore scenarios with scarce calibration data, and evaluated the impact of the size of the calibration set.
> - Small Sample Regime: In Section 5 ("Calibration Set Size") and Figure 4 , we performed a sweep of calibration set sizes from 20 to 200. We found that our method outperforms Self-Consistency with as few as 20 examples (e.g., on WMT EN-DE, MAE drops sharply and stabilizes by 60 examples).
> - Zero-Shot/Few-Shot: While our method is designed for settings where a small validation set exists (a common realistic assumption for production pipelines), the "Few Shot" (FS) baseline we compared against uses those examples in-context. Our results show that using those examples to calibrate the aggregation (Our Method) is far more effective than putting them in the prompt (FS baseline).
>
> [4] (Training vs. Prompting):
>
> We agree that fine-tuning is a valid path to calibration, and we have added the suggested citations regarding training-based methods (as pointed out by reviewer FtNF, and our response under [4] (Training vs. Prompting)) that looked into training methods to improve the LLM-as-a-judge performance. However, in this work, we focus specifically on Inference-Time Compute (ITC). ITC approaches are essential in scenarios where model weights are inaccessible (e.g., using API-based models like GPT-4 or Gemini) or where retraining is prohibitively expensive.
>
> [5] (Transferability):
>
> We have addressed this in our new "Transferability" section and Figure 3.
> - Transfer Success: We found that calibration parameters generally transfer well between tasks with similar "tie regimes." For example, parameters learned on RB2-Math transfer effectively to RB2-Focus and RB2-Safety (see the blue off-diagonal cells in Figure 3).
> - Transfer Limitations: We also identified specific "mismatched regimes." For instance, RB2-Factuality has a very high ground-truth tie rate that the LLM fails to predict naturally. Parameters learned here do not transfer well to tasks where the LLM is naturally better calibrated (like WMT).
> - Quantification: As shown in Appendix E (Table 9), we can quantify this by looking at the learned parameters ($\nu, \gamma$). Tasks in the "High-Correction Regime" transfer well among themselves, but transferring from a "High-Correction" task to a "Low-Correction" task results in performance loss.

---

### Author Response · Authors · 2025-11-26
**Common Response to Reviewers**

We thank the reviewers for their constructive feedback. We have updated the manuscript with new experiments and analyses. The major changes include:

1. Optimization Objective: We transitioned from Maximum Likelihood Estimation (MLE) to minimizing the Discrete Ranked Probability Score (DRPS). This strictly proper scoring rule better captures the ordinal structure of the data. This led to further improvements across some tasks while the performance remained the same for others.

2. Reparametrization of BTD: We combined local and global BTD parameters to use the same BTD model over three hyperparameters $(\beta, \nu, \gamma)$ that could handle both sets of features. This change particularly improved the results for WMT en->de.

3. Transferability: We added new results to the Experiments section to study the transferability of the BTD model across different tasks (See the subsection on Transferability in Section 5 as well as Figure 3).

4. Calibration Efficiency: We included a comprehensive ablation study (Figure 4 in Section 5, Appendix C for full results) demonstrating that our method is data-efficient, stabilizing with as few as 40–60 calibration samples.

---

### Note · Program_Chairs · 2026-01-17
**Submission Desk Rejected by Program Chairs**

The following references in this submission do not refer to real documents and/or have major errors in bibliographic information:

 Han Li, Yuxiang Zhang, Yun Li, Yang Liu, and Cuiyun Dong. Mirror-consistency: Inconsistent minority matters for language model self-consistency. In Findings of the Association for Computational Linguistics: EMNLP 2024, pp. 2364-2377. Association for Computational Linguistics, 2024. URL https://aclanthology.org/2024.findings-emnlp. 135.